# MITIGATING REWARD OVER-OPTIMIZAATION IN DIRECT ALIGNMENT ALGORITHMS WITH ADAPTIVE IMPORTANCE SAMPLING

## ABSTRACT

Recently, Direct Alignment Algorithms (DAAs) such as Direct Preference Optimization (DPO) have emerged as alternatives to the standard Reinforcement learning from human feedback (RLHF) for aligning large language models (LLMs) with human values. Surprisingly, while DAAs do not use a separate proxy reward model as in RLHF, their performance can still deteriorate due to over-optimization – a phenomenon found in RLHF where the policy can exploit failures of the reward model to achieve high rewards but the actual quality of the model begins to degrade. Recent studies find that DAAs tend to increase probability mass on out-of-distribution responses and the training objective in DAAs is heavily underconstrained on these out-of-distribution (OOD) responses due to a mismatch between offline distribution and the LM policy. In this paper, we propose a method to mitigate the distribution shift between the offline distribution and the LM policy by multiplying with an importance weight to reflect the policy distribution. The resulting method, called Adaptive Importance Sampling (AIS), relies on importance sampling techniques and resolves the high variance issue in importance sampling without extra hyper-parameters. Our experiment results showed Adaptive IS can improve win rates by 15% while maintaining lower KL ~~budged~~ budget compared to DAAs.

## 1 INTRODUCTION

Preference learning has emerged as an important part of the fine-tuning process to align large language models with human preference. There are two predominant flavors of preference learning for LLMs. The first approach includes online reinforcement learning from human feedback (RLHF) methods (Ouyang et al., 2022; Christiano et al., 2017). It typically involves a multi-stage procedure: fine-tuning a reward model to capture human preference and fine-tuning the LM policy to maximize the expected reward using online RL algorithms such as Proximal Policy Optimization (Schulman et al., 2017). While empirically performant, this multi-stage procedure is complex and computationally intensive: it requires repeated querying of the reward model as well as sampling from the current policy. A set of alternative methods called direct alignment algorithms (DAAs), avoid fitting separate reward models, instead opting to simply train the policy directly on the offline preference dataset via a ranking loss. The most known examples are Direct Preference Optimization (DPO) (Rafailov et al., 2023), and Identity Preference Optimization (IPO) (Tang et al., 2024c). Since DAAs typically do not sample new responses from the LLM's policy during training, they are characterized as offline preference learning methods.

In RLHF, LMs are trained to optimize a surrogate, imperfect reward function instead of the actual "ground-truth" human reward, resulting in situations where the policy learns to produce responses that achieve high reward scores, but their quality is poor. This phenomenon is often known as the reward over-optimization or reward hacking problem in RLHF (Stiennon et al., 2020b; Ouyang et al., 2022; Chen et al., 2024b; Gao et al., 2022). In the context of direct alignment algorithms (DAAs), reward-hacking-like behaviors still exist even when there is no explicit reward model (Rafailov et al., 2024; Guo et al., 2024). For instance, LLMs fine-tuned with DPO generate responses with increasing length but do not improve the ground-truth win rate (Park et al., 2024a). In another study, Rafailov

et al. (2024) found that DAAs exhibit degradation patterns at various KL-divergence budgets, similar to those in RLHF.

There are several explanations for why the reward over-optimization phenomenon occurs in the classical RLHF pipeline: (1) the reward functions are evaluated on unseen responses and (2) learned reward functions prefer unintended behaviors. Morever, the LLMs can learn to generate OOD examples to exploit these failure modes of RMs (Hendrycks et al., 2021; Rame et al., 2024). Similarly, Rafailov et al. (2024) explains the over-optimization in DAAs by appealing to the under-constrained nature of the optimization problem used in DAAs when extrapolating to OOD samples. As a result, a large amount of extrapolation can potentially be detrimental to the performance of the learned policy.

In this work, we first identify one source of over-optimization in DAAs: the ineffective regularization of DAAs due to the shift between the distribution used for data collection and the trained policy, leading to ineffective use of the KL divergence budget. Our results show that reward over-optimization happens earlier and the performance gain from DAAs diminishes as the offline data shifts away from the LM policy. One approach to mitigate this problem is to add a KL divergence penalty to encourage the model to stay close to reference policy ~~(Song et al., 2024; Fisch et al., 2024)~~ (Song et al., 2024; Fisch et al., 2024; Ding et al., 2024). This additional regularization explicitly prevents the LM policy from pushing a large probability mass to OOD responses. However, these methods are costly since they require repeated sampling from the current policy and are sensitive to hyper-parameters. We propose a novel method based on importance sampling techniques, called Adaptive Importance Sampling (Adaptive IS). Adaptive IS reduces the effects of the distribution shift problem while also balancing the trade-off between bias and variance of the importance ratio to stabilize training. Furthermore, the implementation of Adaptive IS incurs minimal computational overhead, making it highly scalable.

Our main contributions are as follows:

- We study the effect of distribution shift and how it relates to reward over-optimization in DAAs.
- We propose Adaptive Importance Sampling (Adaptive IS), to minimize the distribution gap between offline distribution and the LM policy
- Our results indicate that Adaptive IS outperforms DAAs, with up to a 15% win rate as measured by a golden reward model, while maintaining a lower KL budget.

## 2 PRELIMINARIES

We provide the formulation and background of RLHF and DAAs in sections 2.1 and 2.2, respectively. The over-optimization phenomenon and regularization in DAAs are presented in Section 2.3 and 2.4.

### 2.1 REINFORCEMENT LEARNING FROM HUMAN FEEDBACK (RLHF)

To align LMs with human preferences, the overall RLHF pipelines consist of three stages:

**Supervised Fine-Tuning (SFT)**: Given a pre-trained model and a dataset of prompts $\mathbf{x}$ and response $\mathbf{y}$. Language models are trained for instruction following via maximum-likelihood estimation over next-tokens. The resultant model is then called $\pi_{\text{ref}}(\mathbf{y}|\mathbf{x})$.

**Reward Modeling**: In the second phase, the reference model is prompted with prompts $\mathbf{x}$ to produce pairs of responses $(\mathbf{y_1}, \mathbf{y_2}) \sim \pi_{\text{ref}}(\cdot|\mathbf{x})$. The pair of responses then being labeled by the human to express preferences, which are denoted as $\mathbf{y}^w \succ \mathbf{y}^l|\mathbf{x}$. Typically, user rankings are assumed to follow the Bradley-Terry model:

$$p(\mathbf{y}_1 \succ \mathbf{y}_2|x) = \frac{\exp(r(\mathbf{x}, \mathbf{y}_1))}{\exp(r(\mathbf{x}, \mathbf{y}_1)) + \exp(r(\mathbf{x}, \mathbf{y}_2))} = \sigma\big(r(\mathbf{x}, \mathbf{y}_1) - r(\mathbf{x}, \mathbf{y}_2)\big)$$

This results on preference dataset $\mathcal{D} = \{\mathbf{x}^{(i)}, \mathbf{y}^{w(i)}, \mathbf{y}^{l(i)}\}_{i=1}^N$. We can then use this dataset to train a parametrized reward model $r_\phi(x, y)$ to maximize the differences between $\mathbf{y}^w$ and $\mathbf{y}^l$ using

maximum likelihood estimation with the following objective:

$$\mathcal{L}_R(r_\phi) = -\mathbb{E}_{(\mathbf{x},\mathbf{y}^w,\mathbf{y}^l)\sim\mathcal{D}}[\log\sigma(r_\phi(\mathbf{x},\mathbf{y}^w) - r_\phi(\mathbf{x},\mathbf{y}^l))]$$

**RL Fine-tuning**: After obtaining the learned reward function at the second stage, it can be used to provide feedback to the language model with an on-policy algorithm such as PPO with the following objective:

$$\max_{\pi_\theta} \mathbb{E}_{\mathbf{x}\sim\mathcal{D},\mathbf{y}\sim\pi_\theta(\cdot|\mathbf{x})}\Big[r_\phi(\mathbf{x},\mathbf{y}) - \beta\mathbb{KL}(\pi_\theta||\pi_{\text{ref}})\Big]$$

Where $\beta$ controlling the deviation from the reference policy $\pi_{\text{ref}}$. This constraint prevents the model from deviating too far away from the region that the reward model is well-trained on and prevents mode-collapse to single high-rewards responses.

## 2.2 DIRECT ALIGNMENT ALGORITHMS (DAAs).

While RLHF achieves superior performance in aligning LMs with human preferences, this process is complex and computationally expensive. DAAs address these problems by directly optimizing policy $\pi_\theta$ over preference data. Amongst these algorithms, Direct Preference Optimization is the most popular approach, DPO derived the closed-form solution of Eq 2,

$$\pi^*(\mathbf{y}|\mathbf{x}) = \frac{1}{Z(\mathbf{x})}\pi_{\text{ref}}(\mathbf{y}|\mathbf{x})\exp\left(\frac{1}{\beta}r(\mathbf{x},\mathbf{y})\right)$$

With $Z(\mathbf{x})$ as the normalization function, According to the above equation, we can parameterize the reward function by the log-likelihood ratio between $\pi_\theta$ and $\pi_{\text{ref}}$:

$$r_\theta(\mathbf{x},\mathbf{y}) = \beta\log\frac{\pi_\theta(\mathbf{y}|\mathbf{x})}{\pi_{\text{ref}(\mathbf{y}|\mathbf{x})}} + \beta\log Z(\mathbf{x})$$

This enables us to optimize the LM policy $\pi_\theta$ directly with human feedback data:

$$\mathcal{L}_{\text{DAA}}(\pi_\theta,\pi_{\text{ref}}) = \mathbb{E}_{\mathbf{x}\sim\mathcal{D},(\mathbf{y}^w,\mathbf{y}^l)}\sim\pi_{\text{ref}}(\cdot|\mathbf{x})\Big[f\Big(\beta\log\frac{\pi_\theta(\mathbf{y}^w|\mathbf{x})}{\pi_{\text{ref}}(\mathbf{y}^w|\mathbf{x})} - \beta\log\frac{\pi_\theta(\mathbf{y}^l|\mathbf{x})}{\pi_{\text{ref}}(\mathbf{y}^l|\mathbf{x})}\Big)\Big]$$

Where $f$ is a convex loss function. When $f(x) = -\log\sigma(x)$, we recover standard DPO objective (Rafailov et al., 2023), other popular objectives include: IPO (Azar et al., 2024) with $f(x) = (x-1)^2$. Other objectives can be found in (Tang et al., 2024c). In this paper, we will focus on these 2 standard objectives due to limited computational resources.

## 2.3 OVER-OPTIMIZATION IN DAAS

Gao et al. (2022) refer to the *over-optimization* phenomenon as the situation where algorithms consume a large *optimization budget* without improving or even reducing performance. In this study, the KL divergence $\mathbb{KL}(\pi_\theta,\pi_{\text{ref}})$ is used as an optimization budget since it measures how far the optimized policy $\pi_\theta$ drifts away from the reference policy $\pi_{\text{ref}}$ during training. Rafailov et al. (2024) study the trade-off between KL divergence and policy performance under three direct alignment objectives DPO, IPO, and SLiC. They observe clear over-optimization after a certain time during training when an additional increase in the KL budget leads to decreasing model performance. This pattern persists across model sizes, and smaller models often exhibit clearer signs of over-optimization. Moreover, regularization methods such as length regularization can not mitigate this problem. Tang et al. (2024a) observe that both online and offline variants of DAAs suffer from over-optimization, however, online achieve better budget and performance trade-offs than offline. It's not clear why since both of them are bottlenecked by an offline pairwise preference dataset.

## 2.4 REGULARIZATION IN DAAS

In this section, we borrow analysis from GPO Tang et al. (2024d) to investigate the regularization effect of DAAs' loss functions. We first denote the log ratio difference as $\rho_\theta := \log\frac{\pi_\theta(\mathbf{y}^w)}{\pi_{\text{ref}}(\mathbf{y}^w)} - \log\frac{\pi_\theta(\mathbf{y}^l)}{\pi_{\text{ref}}(\mathbf{y}^l)}$, then the DAA loss can be written as the following:

$$\mathcal{L}_{\text{DAA}}(\rho_\theta) = \mathbb{E}_{\mathbf{x}}\mathbb{E}_{(\mathbf{y}^w,\mathbf{y}^l)\sim\pi_{\text{ref}}}[f(\beta\rho_\theta)].$$

We consider the Taylor expansion around $\rho_\theta = 0$,

$$\underbrace{\mathbb{E}_{\mathbf{x}}\mathbb{E}_{(\mathbf{y}^w,\mathbf{y}^l)\sim\pi_{\text{ref}}}\left[f\left(\beta\rho_\theta\right)\right]}_{\text{offline loss}} \approx f(0) + \underbrace{f'(0)\beta\cdot\mathbb{E}_{\mathbf{x}}\mathbb{E}_{(\mathbf{y}^w,\mathbf{y}^l)\sim\pi_{\text{ref}}}\left[\rho_\theta\right]}_{\text{preference optimization}} + \underbrace{\frac{f''(0)\beta^2}{2}\cdot\mathbb{E}_{\mathbf{x}}\mathbb{E}_{(\mathbf{y}^w,\mathbf{y}^l)\sim\pi_{\text{ref}}}\left[\rho_\theta^2\right]}_{\mu\text{-weighted squared loss}},$$

$$(1)$$

Consider the expectation of gradient of the $\mu$-weighted squared loss term,

$$\mathbb{E}_{\mathbf{x}}\mathbb{E}_{\mathbf{y}\sim\pi_{\text{ref}}}\left[\nabla_\theta\frac{1}{2}\rho_\theta^2\right].$$

Tang et al. (2024d) show that if $\mu = \pi_\theta$ then this expectation will recover the update of reverse KL regularization, i.e.

$$\mathbb{E}_{\mathbf{x}}\mathbb{E}_{\mathbf{y}\sim\pi_\theta}\left[\nabla_\theta\frac{1}{2}\rho_\theta^2\right] = C\mathbb{E}_{\mathbf{x}}\nabla_\theta\mathbb{KL}\left(\pi_\theta,\pi_{\text{ref}}\right) \tag{2}$$

where $C$ is constant depended on $\beta$, $f'(0)$ and $f''(0)$. This equality suggests that DAAs enforce regularization via optimizing a $\mu$-weigthed objective.

Note that the approximation in Eq. 1 is only valid when $\rho_\theta$ is small and Eq. 2 is only valid when the expected gradient under current policy $\pi_\theta$ can be estimated using training data. These conditions are generally not held when the training progresses. As a result, the algorithms can not guarantee bounded reverse KL if the training data does not cover the response space well (Song et al., 2024). In section 3.2, we provide an analysis of the regularization effect in DAAs using a didactic setting.

## 3 METHODOLOGY

### 3.1 ADAPTIVE IMPORTANCE SAMPLING (ADAPTIVE-IS)

In the DAAs algorithm, human preference data does not need to be collected from the starting policy $\pi_{\text{ref}}$. Moreover, even if $\mu = \pi_{\text{ref}}$, during training DAAs tend to assign a high probability mass on OOD responses that are not presented in offline data (Tajwar et al., 2024; Rafailov et al., 2024). Once the policy $\pi_\theta$ moves far away from $\pi_{\text{ref}}$, this can potentially be detrimental to performance and offline data may not have sufficient coverage to rectify.

To mitigate this problem, a simple approach is to apply online sampling training to collect responses from the current policy $\pi_\theta$ and use an external reward to correct these biases from the LM policy ~~(Guo et al., 2024)~~(Calandriello et al., 2024; Guo et al., 2024).

$$\mathcal{L}_{\text{Online-DPO}}(\pi_\theta, \pi_{\text{ref}}) = -\mathbb{E}_{\mathbf{x}\sim\mathcal{D},(\mathbf{y}^w,\mathbf{y}^l)\sim\pi_\theta(\cdot|\mathbf{x})}\left[\log\sigma\left(\beta\log\frac{\pi_\theta(\mathbf{y}^w|\mathbf{x})}{\pi_{\text{ref}}(\mathbf{y}^w|\mathbf{x})} - \beta\log\frac{\pi_\theta(\mathbf{y}^l|\mathbf{x})}{\pi_{\text{ref}}(\mathbf{y}^l|\mathbf{x})}\right)\right]$$

However, online training is considerably more complex than off-policy methods, involving multi-stage training (requiring training an external reward) and sampling from the LM policy during training, incurring significant computational costs.

Our method aims to minimize the distribution gap between offline distribution and the policy distribution while does not need online sampling using importance sampling, a technique to estimate expectations under one distribution given samples from a reference distribution $\pi_{\text{ref}}$, which leads to an unbiased estimation of online-DPO objective :

$$\mathcal{L}_{\text{IS-DPO}}(\pi_\theta, \pi_{\text{ref}})$$

$$= -\mathbb{E}_{\mathbf{x}\sim\mathcal{D},\mathbf{y}^w,\mathbf{y}^l\sim\pi_{\text{ref}}(\cdot|\mathbf{x})}\left[\left(w(\mathbf{x},\mathbf{y}^w,\mathbf{y}^l)\log\sigma\left(\beta\log\frac{\pi_\theta(\mathbf{y}^w|\mathbf{x})}{\pi_{\text{ref}}(\mathbf{y}^w|\mathbf{x})} - \beta\log\frac{\pi_\theta(\mathbf{y}^l|\mathbf{x})}{\pi_{\text{ref}}(\mathbf{y}^l|\mathbf{x})}\right)\right)\right]$$

where the importance weights $w(\mathbf{x},\mathbf{y}^w,\mathbf{y}^l) = \frac{\pi_\theta(\mathbf{y}^w|\mathbf{x})}{\pi_{\text{ref}}(\mathbf{y}^w|\mathbf{x})}\frac{\pi_\theta(\mathbf{y}^l|\mathbf{x})}{\pi_{\text{ref}}(\mathbf{y}^l|\mathbf{x})}$. Here, the importance weight is the ratio of sequence-level probability between $\pi_\theta$ and $\pi_{\text{ref}}$, e.g. $\frac{\pi_\theta(\mathbf{y}|\mathbf{x})}{\pi_{\text{ref}}(\mathbf{y}|\mathbf{x})} = \prod_{t=1}^T\frac{\pi_\theta(y_t|\mathbf{x},\mathbf{y}_{<t})}{\pi_{\text{ref}}(y_t|\mathbf{x},\mathbf{y}_{<t})}$. The update is multiplied by this importance weight to adjust the action probabilities so that the expectation is as if the actions were sampled according to the LM policy $\pi_\theta$.

**Adaptive Importance Sampling**   Direct computing the importance weights in training can suffer from extremely high variance when $\pi_\theta$ deviate far away from $\pi_{\text{ref}}$. To mitigate this problem, we consider another estimator, called *Exponential Smoothing Importance Sampling* (Aouali et al., 2023; Korba & Portier, 2022), which is defined as:

$$\mathcal{L}_{\text{DPO}}(\pi_\theta, \pi_{\text{ref}})$$

$$= -\mathbb{E}_{(\mathbf{x}, \mathbf{y}^w, \mathbf{y}^l) \sim \mathcal{D}} \left[ \left( \frac{\pi_\theta(\mathbf{y}^w|\mathbf{x})}{\pi_{\text{ref}}(\mathbf{y}^w|\mathbf{x})} \frac{\pi_\theta(\mathbf{y}^l|\mathbf{x})}{\pi_{\text{ref}}(\mathbf{y}^l|\mathbf{x})} \right)^\alpha \log \sigma \left( \beta \log \frac{\pi_\theta(\mathbf{y}^l|\mathbf{x})}{\pi_{\text{ref}}(\mathbf{y}^w|\mathbf{x})} - \beta \log \frac{\pi_\theta(\mathbf{y}^l|\mathbf{x})}{\pi_{\text{ref}}(\mathbf{y}^l|\mathbf{x})} \right) \right]$$

where $\alpha$ serve as a regularization to trade-offs between bias and variance of the Importance weight estimator. It is easy to see that when $\alpha = 0$, we recover DPO loss and when $\alpha = 1$, we obtain DPO with importance sampling. We give further details on how $\alpha$ trade-off between bias and variance in Appendix A

**How to choose $\alpha$?** Given the LMs $\pi_\theta$ is an auto-regressive model. Where for each prompt $\mathbf{x}$, The LM $\pi_\theta$ generate $\mathbf{y}$ in an auto-regressive manner:

$$\pi_\theta(\mathbf{y}|\mathbf{x}) = \prod_{t=1}^{T} \pi_\theta(y_t, (\mathbf{x}, \mathbf{y}_{<t})$$

As the number of the tokens $T$ increases, the variance of the importance weight can grow exponentially with respect to the number of tokens. Thus, we should decrease $\alpha$ value when the number of tokens is large and vice versa. by setting $\alpha = \frac{1}{|y|}$, we can adaptively trade-offs between bias and variance of importance weight. A detailed analysis of the variance and the effect of $\alpha$ are given in Section E, Appendix.

## 3.2   An analysis of regularization effect in DAAs and Adaptive-IS DAAs

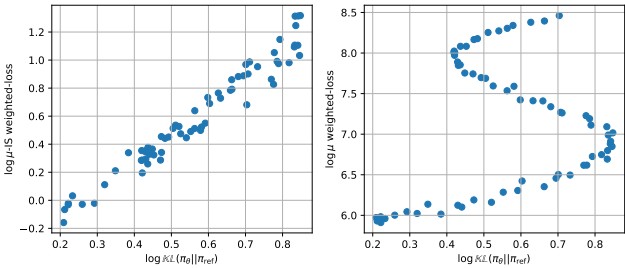

Figure 1: Correlation between KL divergence and $\mu$-weighted loss and $\mu$-IS weighted loss. We can see that $\mu$-IS weighted loss achieve high correlation with the KL divergence.

As the section 2.4 has shown the square loss term in DAAs only serves as a local approximation of KL divergence when $\pi_\theta$ is near $\pi_{\text{ref}}$, as the LM policy $\pi_\theta$ deviates far away from the reference model, the correlation between 2 objectives becomes more difficult to grasp. To see how importance sampling can enforce a more effective regularization in DAAs, we experiment with a synthetic setup using a Mixture of Gaussian and measure the correlation between $\mu$-weighted loss and $\mu$-weighted loss with importance sampling ($\mu$-IS weighted loss).

$$\mathbb{E}_{\mathbf{x} \sim \mathcal{D}, (\mathbf{y}^w, \mathbf{y}^l) \sim \mu(\cdot|\mathbf{x})} \left[ \frac{w(\mathbf{x}, \mathbf{y}^w, \mathbf{y}^l)}{2} \left( \log \frac{\pi_\theta(\mathbf{y}^w|\mathbf{x})}{\pi_{\text{ref}}(\mathbf{y}^w|\mathbf{x})} - \log \frac{\pi_\theta(\mathbf{y}^l|\mathbf{x})}{\pi_{\text{ref}}(\mathbf{y}^l|\mathbf{x})} \right)^2 \right]$$

The offline distribution $\mu$ is parameterized as $\mu = \frac{3}{10}\mathcal{N}(-0.8, 0.2^2) + \frac{4}{10}\mathcal{N}(0, 0.2^2) + \frac{3}{10}\mathcal{N}(0.8, 0.2^2)$. We assume $\pi_{\text{ref}} = \mu$ and the policy distribution $\pi_\theta = \mathcal{N}(\theta, 0.1^2)$, where $\theta$ is a parameter, we varying $\theta$ from $[-1, 1]$ and estimate KL divergence, $\mu$-weighted loss and $\mu$-IS weighted loss, we generate 2000 samples to estimate these objectives.

In figure 1, we show the correlation between the KL divergence and $\mu$-weighted loss and $\mu$-IS weighted loss under log scale with $\theta$ varying from $[-1, 1]$. When $\pi_\theta$ is close to $\pi_{\text{ref}}$, both 2 losses

exhibit a high correlation with KL divergence. But for $\mu$-weigthed loss, the correlation starts to break down when $\pi_\theta$ moves far away from $\pi_{\text{ref}}$, while $\mu$-IS still exhibits a high correlation with KL divergence.

# 4 EXPERIMENTS

In this section, we will first examine how distribution shift affects the performance of DAAs. We find that under distribution shift, the performance gain from DAAs is margin and decreases when the offline data shifts away from the LM policy. Moreover, reward-overoptimization happen faster under distribution shift.

Then, we will evaluate our methods in standard RLHF datasets: TL;DR summarization.

## 4.1 EXPERIMENT SETTINGS

We adopt a synthetic setup from Gao et al. (2022) to study the trade-off between KL divergence and policy performance. We first train a *golden* reward model with a Pythia-6.9b from the initial human preference dataset and then use it to label preference data for training offline algorithms. The gold reward model will be much larger than the optimized policy to simulate the complexity of human preferences for the LM policy to be captured given a finite dataset.

**Dataset**: For all experiments, we will use Reddit TL;DR summarization dataset Stiennon et al. (2020a)(Stiennon et al., 2020a). It is a summarization dataset with SFT split, consisting of 116,722 human-written summaries and preference split, comprising 92,858 human-annotated preference pairs.

**Pretrained Model**: All of our experiments will be carried out using the Pythia family of Large Language Models Biderman et al. (2023) (Biderman et al., 2023) with 1B model sizes due to limited computational resources. All models have gone through supervised fine-tuning on the SFT split of the TL;DR dataset, resulting $\pi_{\text{ref}}$ policy. The model is then trained on preference learning data for 1 epoch using AdamW optimizer, with a cosine decay schedule and a learning rate of $1e - 6$.

**Model Evaluation**: We evaluate the performance of any policy by the win rate against 512 reference summaries available in the SFT split. The golden reward model determines the win rate. We evaluate with 2 standard objectives in DAAs: DPO and IPO. Following previous works, we use the KL divergence between the current policy $\pi_\theta$ and the reference policy $\pi_{\text{ref}}$ as a measure of optimization budget (Rafailov et al., 2024; Tang et al., 2024a; Gao et al., 2022).

## 4.2 MAIN RESULTS

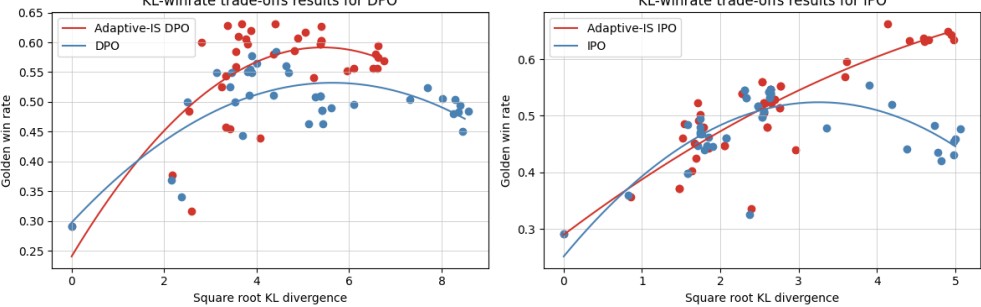

Figure 2: Trade-off between performance and KL divergence for DPO and Adaptive IS with varying regularization strength. We see that Adaptive IS achieves superior performance and KL efficiency.

In this section, we evaluate the over-optimization phenomenon when using Adaptive IS and compare it against two baselines: DPO and IPO. Our key findings are illustrated in Figure 2, which displays the model win rates using an evaluation set of prompts judged by the golden reward model. It's important to note that over-optimization for DAAs occurs when the performance shows a hump-shaped

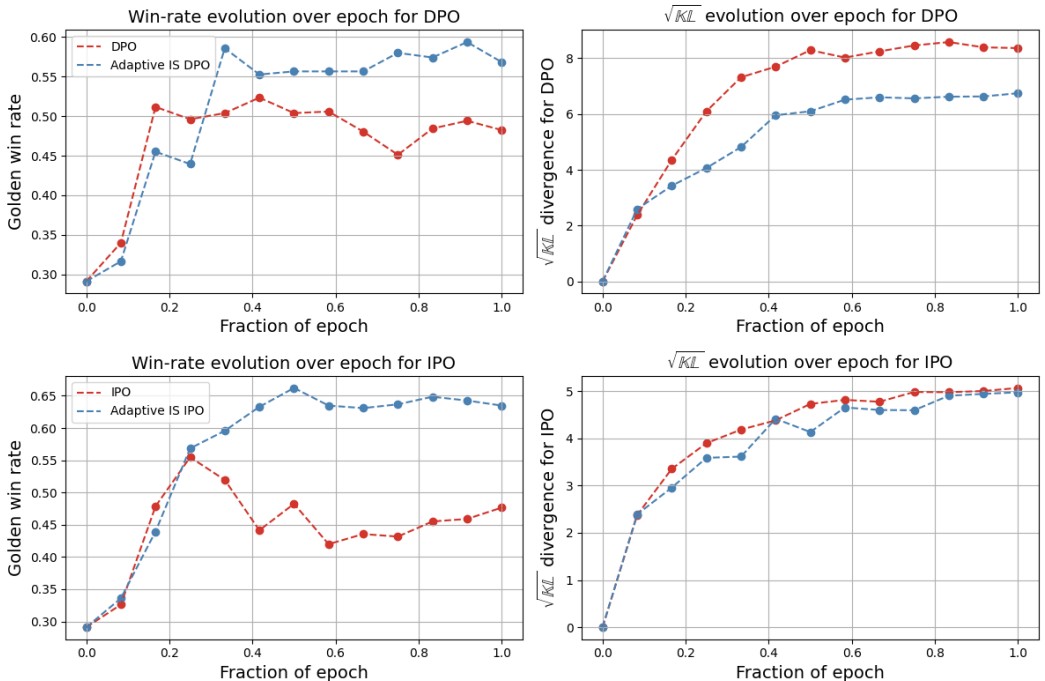

Figure 3: Evolution of win-rates, and KL divergence. Adaptive IS model achieves higher final win rate over standard DPO model with less than 35% of the KL budget Moreover, Adaptive IS maintains consistent performance throughout training, while standard DPO performance peaks early at 20% of the first epoch and start to decreasing performance.

pattern, where increasing the KL budget leads to a decrease in model performance (Rafailov et al., 2024). These patterns can be clearly observed from the DPO and IPO tradeoff curves. On the other hand, Adaptive IS-DPO (resp IPO) outperforms standard DPO (resp IPO) by a large margin given a smaller KL budget, increasing performance by over 10% under the same KL budget. The results demonstrate that Adaptive IS can address the over-optimization issue and uses the KL divergence budget more efficiently than offline.

In previous studies, it has been shown that DAAs tend to show early convergence behavior during training. They achieve their highest performance after being trained on only a small portion of the data. Subsequently, their performance starts decreasing in conjunction with a rise in KL divergence metrics (Park et al., 2024a; Rafailov et al., 2023). In figure 3, we analyze the intra-epoch training dynamics patterns of standard DPO, IPO, and the AIS variants as configurations with $\beta = 0.01$. After 20% of the epoch, DPO has reach it highest win-rate and start to descend while increasing KL steadily with further training. In contrast, Adaptive IS-DPO shows no degradation as the training progresses and achieves higher final win rates. This can be explained that at the initial steps, DAAs objective always initialized as the reference model, the offline data distribution is similar to the LM policy distribution and can make a solid improvement, as the LM policy deviates far away from the reference model. Offline algorithms become less effective as they no longer represent the distribution encountered during on-policy. This growing discrepancy between the training and test time may lead to sub-optimal performance. In contrast, AIS can leverage pre-collected data and select training instances that benefit the learning process.

### 4.3 HOW DOES DISTRIBUTION SHIFT AFFECT OFFLINE PREFERENCE OPTIMIZATION?

Previous works often attribute the sub-optimal performance of DAAs to a distribution gap between the current policy and the policy used to sample training data. In this experiment, we create a setting where we can control the gap between the training data and the training policy and try to observe the effect of these gaps on the final performance. Specifically, we first perform DPO fine-tuning on the

SFT model $\pi_{\text{ref}}$ and collect two checkpoints $\pi_{\theta_1}$ and $\pi_{\theta_2}$, ordered by number of training iterations. Then we generate pairs of responses using $\pi_{\text{ref}}, \pi_{\theta_1}$ and $\pi_{\theta_2}$, resulting to three synthetic datasets $D_1$, $D_2$ and $D_3$, respectively. These datasets are labeled using the golden preference model. By following this procedure, $D_1$, $D_2$ and $D_3$ are gradually shifted away form $\pi_{\text{ref}}$. We then finetune 3 LM policies initialized from $\pi_{\text{ref}}$ on these 3 datasets using DPO objective with varying regularization strength.

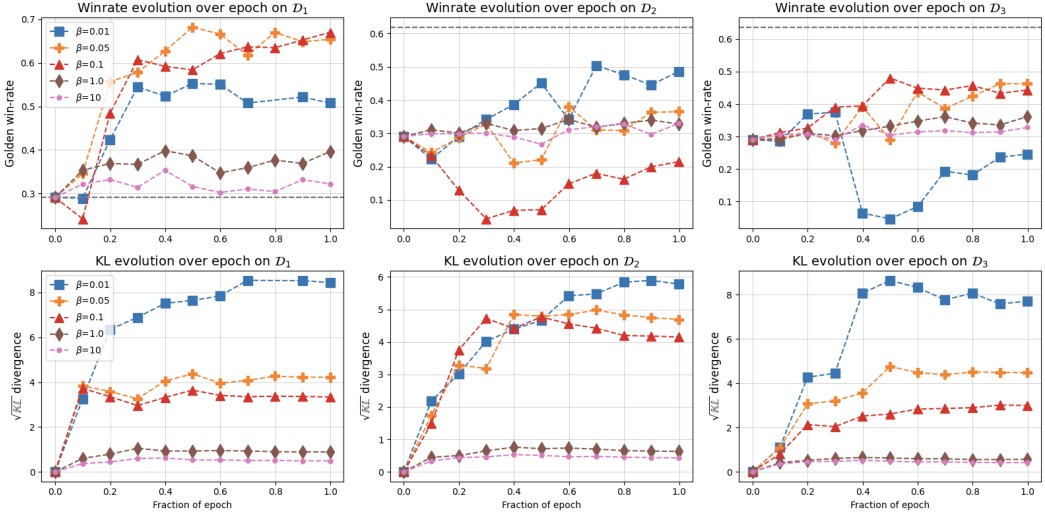

Figure 4: Win-rate and KL divergence against the fraction of epoch results for datasets $\mathcal{D}_1, \mathcal{D}_2, \mathcal{D}_3$ gradually shift away from $\pi_{\text{ref}}$

.

In figure 4, we observed the same phenomenon as in (Tang et al., 2024a) where the SFT data achieves the best performance compared to the other 2 datasets. For the other 2 datasets, the performance gain is margin and cannot achieve the same level of performance of the data generated policy. Moreover, as the data gradually shifts away from the initial policy, the performance gain from offline preference learning becomes less significant. In figure 5 (left), we report the win-rate KL trade-off between the

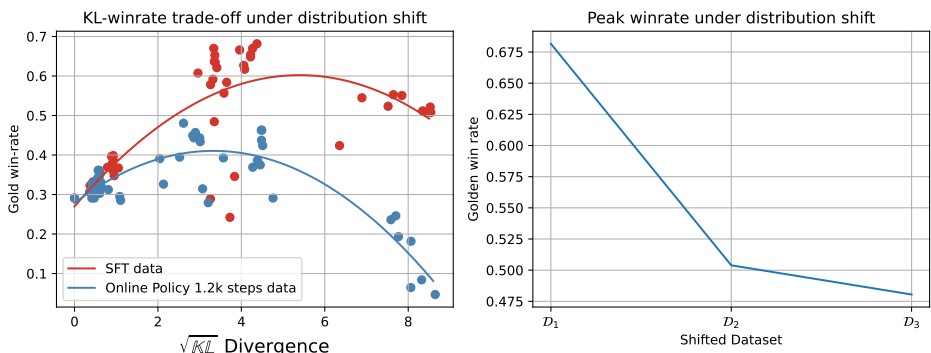

Figure 5: Left: KL-win rate tradeoff results under distribution shift. Reward-over-optimization happens earlier under distribution shift and cannot achieve satisfying performance even though the online policy data performs much better than the SFT model. Right: Peak win rate across different datasets. As the data gradually shifts, the performance of DAAs starts to degrade.

policy learned from SFT data and data $\mathcal{D}_3$ that generated from the policy with the highest win rate. We can see that reward-over-optimization happens earlier under distribution shift than the SFT data and cannot achieve satisfying performance even though the offline is generated from a higher win-rate policy, showing the importance of how different the offline distribution is to the LM policycan have large effect to the performance of DAAs. In figure 5 (right), we show the peak win rate of

various regularization strengths across different shifted datasets. As the offline shift aways from the initial LM policy $\pi_\theta$, the performance gain from DAAs methods starts to decrease.

## 5 RELATED WORKS

**Preference fine-tuning.**   There are two main approaches for fine-tuning language models based on user preferences. The first approach involves online reinforcement learning methods such as RLHF (Ouyang et al., 2022). This method includes multiple steps: fine-tuning a reward model to capture preferences and optimizing language models to maximize the reward scores. The second approach, known as direct alignment algorithms (DAAs), aims to simplify the multi-step process of RLHF. DAAs directly update the language model's policy using human feedback. Examples of DAAs include Direct Preference Optimization (DPO) (Rafailov et al., 2023), and Identity Preference Optimization (IPO) (Tang et al., 2024c). Since DAAs don't typically generate new responses from the language model's policy during training, they are considered offline preference learning methods.

**Reward-Overoptimization in RLHF.**   Gao et al. (2022) refer to the *over-optimization* phenomenon as optimizing too much against a surrogate objective eventually hinders the true objective. They introduce a synthetic setup to study the trade-off between the KL divergence $\mathbb{KL}(\pi_\theta, \pi_{\text{ref}})$ and the policy performance. In the context of RLHF, prior works have observed that while the LLM's expected reward increases the actual quality of the model's output decreases. This phenomenon is termed reward exploitation or reward "over-optimization" in RLHF and relates to problems such as verbosity bias. Many works try to address this problem by improve robustness of the reward model: (Shen et al., 2023) proposed to use a smaller reward model to capture length bias and use a larger reward model to learn true reward. (Coste et al., 2024) using an ensemble of rewards improves OOD robustness, (Rame et al., 2024) use weighted-averaged reward models. While these methods have been shown to effectively mitigate reward-overoptimization. Reward-overoptimization in DAAs does not train a reward model, so previous approaches cannot be directly applied to this setting.

**Over-optimization in DAAs.**   Recent works have shown that DAAs also exhibit reward "over-optimization" behavior such as length bias (Park et al., 2024a). Unlike standard RLHF, these offline algorithms do not train an explicit reward function but directly finetune the LMs. However, research addressing over-optimization in offline learning is still limited compared to standard RLHF. Rafailov et al. (2024) explains why over-optimization occurs by pointing to the under-constrained nature of the optimization problem used in DAAs. Park et al. (2024a) try to tackle this problem using reward shaping to eliminate verbosity bias.

**Performance gap between online and offline alignment.**   In this work, we draw the relationship between reward-overoptimization problems in offline alignment algorithms and distribution issues of shift in offline reinforcement learning context (Levine et al., 2020; Kumar et al., 2020). That is, during training, LMs $\pi_\theta$ is trained on data that is generated from reference model $\pi_{\text{ref}}$. However, during deployment, it will be queried on its own distribution. which may lead to performance degradation if the LMs are very unlikely to visit states that are present in the offline data Chen et al. (2024a). The most closely related to our works is that of (Zhou et al., 2024), where they try to minimize the distribution gap between offline and the LM policy simulating on-policy learning with off-policy preference data where they approximate the importance weight by a constant instead of using reference probability. However, they do not provide an explanation for how using length-normalization helps in balancing the trade-offs between bias and variance in the importance weight.

## 6 CONLUSION

We study the problem of reward-overoptimization in Direct Alignment Algorithms (DAAs). We showed that one of the main sources in reward-overoptimization in DAAs is due to the mismatch between offline distribution and the LM policy. To reduce this distribution gap problem, we introduce Adaptive Importance Sampling (Adaptive IS), a technique to estimate samples under the LM policy distribution given samples from the offline distribution while resolving the high variance issue of the importance ratio estimation. Our results showed that Adaptive IS improves performance and is highly effective at combating reward over-optimization in DAAs.

**Limitation.** In this paper, we adopt the synthetic setup used by (Gao et al., 2022), where we assume the golden reward model as the ground truth reward. However, this golden reward model may not accurately represent real-world human preferences. Moreover, we did not experiment with larger models and other datasets due to limited computational resources.

Another limitation is that we assume that the preference data is generated by the reference model. Which is not always hold in practice. In most cases, the preference dataset is sampled from an unknown policy $\mu$, we can only estimate this policy using maximum likelihood estimation, which results in the reference model $\pi_{\text{ref}}$.

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

## A  BIAS-VARIANCE TRADE-OFF OF EXPONENTIAL IMPORTANCE SAMPLING

Given a prompt $\mathbf{x}$, we have that $\mathbb{E}_{y \sim \pi_{\mathrm{ref}}(\cdot|\mathbf{x})}\left[\frac{\pi_\theta(\mathbf{y}|\mathbf{x})}{\pi_{\mathrm{ref}}(\mathbf{y}|\mathbf{x})}\right] = 1$. For a given $\alpha \in [0,1]$. The bias of Adaptive Importance Sampling is

$$\mathbb{E}_{y \sim \pi_{\mathrm{ref}}(\cdot|\mathbf{x})}\left[\left(\frac{\pi_\theta(\mathbf{y}|\mathbf{x})}{\pi_{\mathrm{ref}}(\mathbf{y}|\mathbf{x})}\right)^\alpha\right] = \sum_y \pi_{\mathrm{ref}}(\mathbf{y}|\mathbf{x})\left(\frac{\pi_\theta(\mathbf{y}|\mathbf{x})}{\pi_{\mathrm{ref}}(\mathbf{y}|\mathbf{x})}\right)^\alpha$$

$$\leq \left(\sum_y \pi_{\mathrm{ref}}(\mathbf{y}|\mathbf{x})\left(\frac{\pi_\theta(\mathbf{y}|\mathbf{x})}{\pi_{\mathrm{ref}}(\mathbf{y}|\mathbf{x})}\right)\right)^\alpha \quad (\text{Jensen Inequality})$$

$$\leq 1$$

For the variance of Adaptive Importance Sampling, we have

$$\mathbb{V}ar\left[\left(\frac{\pi_\theta(\mathbf{y}|\mathbf{x})}{\pi_{\mathrm{ref}}(\mathbf{y}|\mathbf{x})}\right)^\alpha\right] \leq \mathbb{V}ar\left[\left(\frac{\pi_\theta(\mathbf{y}|\mathbf{x})}{\pi_{\mathrm{ref}}(\mathbf{y}|\mathbf{x})}\right)^\alpha\right] + \left(\mathbb{E}\left[\left(\frac{\pi_\theta(\mathbf{y}|\mathbf{x})}{\pi_{\mathrm{ref}}(\mathbf{y}|\mathbf{x})}\right)^\alpha - 1\right]\right)^2$$

$$= \mathbb{E}\left[\left(\left(\frac{\pi_\theta(\mathbf{y}|\mathbf{x}}{\pi_{\mathrm{ref}}(\mathbf{y}|\mathbf{x})} \frac{\pi_\theta(\mathbf{y}|\mathbf{x})}{\pi_{\mathrm{ref}}(\mathbf{y}|\mathbf{x})}\right)^\alpha - 1\right)^2\right]$$

$$\leq \mathbb{E}\left[\left(\frac{\pi_\theta(\mathbf{y}|\mathbf{x})}{\pi_\theta(\mathbf{y}|\mathbf{x}} {}_\alpha \frac{\pi_\theta(\mathbf{y}|\mathbf{x})}{\pi_{\mathrm{ref}}(\mathbf{y}|\mathbf{x})} - 1\right)^2\right] \leq \mathbb{E}\frac{\pi_\theta(\mathbf{y}|\mathbf{x}}{\pi_{\mathrm{ref}}(\mathbf{y}|\mathbf{x}} - 1^2$$

## B  KL DIVERGENCE AND GOLDEN WIN-RATE CALCULATION

~~We calculate KL divergence on the full distribution over next token under~~ The calculation of KL divergence in our experiments is based on (Tang et al., 2024b) where the KL is estimated by taking on-policy samples under the current LM $\pi_\theta$ ~~and $\pi_{\mathrm{ref}}$. Therefore, we caleulcate KL divergence acording to (Tang et al., 2024a). Specifically, given a response consists of $T$ tokens. For each partial completion, we can calculate the distribution over the next tokens of both~~. Specifically, we first sample $N$ input prompts $\{\mathbf{x}_i\}_{i=1}^N$ from the evaluation set. For each input prompt $\mathbf{x}_i$, we generate a response $\mathbf{y}_i$ using the current policy $\pi_\theta$. Let $T_i$ be the length of the response $\mathbf{y}_i$, we compute the KL divergence between $\pi_\theta$ and $\pi_{\mathrm{ref}}$ ~~.~~ as follows:

~~The KL divergence will be calculated for each time step $i$, this results an unbiased estimate of KL divergence:~~

$$\frac{1}{N}\sum_{n=1}^N \sum_{t=1}^T \mathbb{KL}\left(\pi_\theta(\cdot|\mathbf{x}, \mathbf{y}_{<t}), \pi_{\mathrm{ref}}(\cdot|\mathbf{x}, \mathbf{y}_{<t})\right)$$

~~Where $N$ is number of samples in the evaluation set.~~

We set $N = 512$ in our experiments.

For Gold win-rate calculation, we first use a well fine-tuned pythia 6.9b in (Huang et al., 2024). The model achieve $\approx 70\%$ accuracy in evaluation set and achieving 76.7

For a given prompt $x$, we first sample a response $y \sim \pi_\theta(\cdot|x)$ and then use the golden reward model $r^{\mathrm{gold}}$ to compare against reference summaries $y_{\mathrm{ref}}$ in evaluation set to determine the win-rate with the following calculation:

$$\frac{1}{N}\sum_{i=1}^N \mathbb{1}\{r^{\mathrm{gold}}(x, y) > r^{\mathrm{gold}}(x, y_{\mathrm{ref}})\}$$

## C  COMPARISON WITH ONLINE ALIGNMENT METHODS

We conducted further experiments where we compared DAAs with online alignment methods. We consider REINFORCE Leave-One-Out (RLOO) (Ahmadian et al., 2024). For RM training, we use a learning rate of $3 \times 10^{-6}$ with a batch size of 64 and a cosine learning scheduler. The reward model is trained on the preference dataset that is labeled from the golden reward model. We RL fine tuning, use a batch size of 512 and the number of generated samples per prompt $k$ is set to 2, we train it for 1200 steps, resulting in approximately 3.3 epochs with a learning rate of $3 \times 10^{-6}$, and a constant linear scheduler with a warm-up ratio of 3%. We present the result in Figure 6.

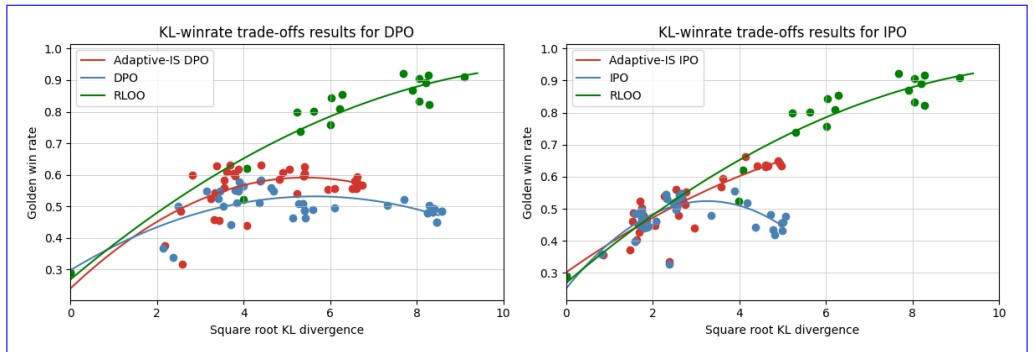

Figure 6: Trade-off between performance and KL divergence for alignment methods with varying regularization strength.

As expected, RLOO achieves a better win rate compared to DPO and AIS-DPO and utilizes a better KL budget. The result also shows that AIS-DPO helps close the gap between offline and online algorithms

## D  EXPERIMENTAL DETAILS

We follow the codebase from the N+ implementation of RLHF (Huang et al., 2024) ~~with default hyper-parameters as shown in the tables below : SFT hyperparameters.~~ **Parameter** . We use `transformers` (Wolf et al., 2020) library implementation of Pythia models in conjunction with `deepspeed` ZERO Stage 2 Rasley et al. (2020). All models are quantized to `bfloat16` dtype. We provide additional details on our training and data preprocessing below

**Data-preprocessing**: We follow data-preprocessing process from (Huang et al., 2024). We truncate the prompt to a maximum of 512 tokens, where the truncation is only applied at the paragraph level. All input strings will be formatted with the following template:

`SUBREDDIT: r/{subreddit}\n\nTITLE: {title}\n\nPOST: {post}\n\nTL;DR:`

~~Value~~**SFT Training** ~~Learning rate 3e-6 Epochs 1 Batch size~~ We use the SFT split, which contains an input query and a reference summary written by humans. We use a learning rate of $3 \times 10^{-6}$ and a batch size of 64 with gradient accumulation steps of 8. We do not apply warm-up ~~steps 0DPO hyperparameters.~~ **Parameter Value** ~~Learning rate 1e-6 Epochs 1 Batch size~~ and train for one epoch. **Preference Training**: We train preference algorithms using the initialized SFT Pythia models. We train for 1450 steps with a learning rate of $1 \times 10^{-6}$ with a batch size of 64 ~~Warm-up steps~~ with gradient accumulation steps of 8, we use a cosine learning rate scheduler with 150 warm-up steps. ~~Generation hyperparameters.~~ **Parameter Value** ~~Max prompt length 512 Max new tokens 128 Temperature 0.01~~

## E  THE NECESSITY OF THE ADAPTIVE HEURISTIC

Since we are working with an auto-regressive language model, the importance weights are computed as the product of the importance ratio of many timesteps. Let $T$ be the length of the response $\mathbf{y}$ to

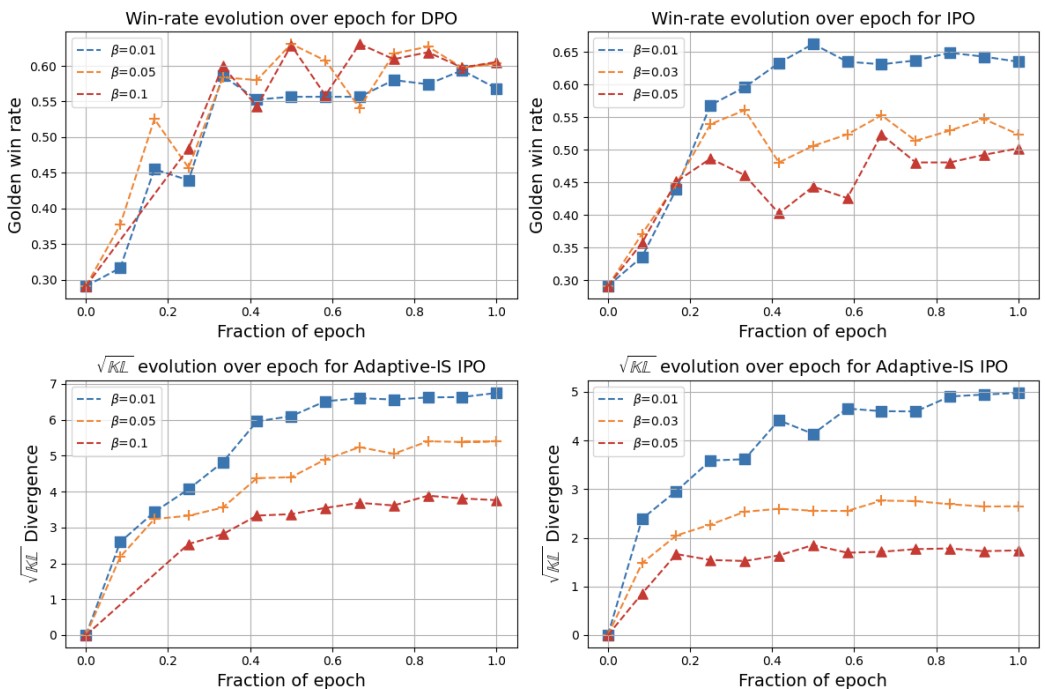

Figure 7: Result on intra-epoch training dynamics of Adaptive IS. The top row shows win-rate against fraction on an epoch and the bottom row shows the Square root of KL evolution. Adatpive IS maintains consistent performance throughout the training process.

an input prompt $\mathbf{x}$, the importance weight is calculated based on the following equation

$$w(\mathbf{x}, \mathbf{y}) = \frac{\pi_\theta(\mathbf{y}|\mathbf{x})}{\pi_{\text{ref}}(\mathbf{y}|\mathbf{x})} = \prod_{t=1}^{T} \frac{\pi_\theta(\mathbf{y}_t|\mathbf{x}, \mathbf{y}_{<t})}{\pi_{\text{ref}}(\mathbf{y}_t|\mathbf{x}, \mathbf{y}_{<t})}$$

Thus, the variance of the IS estimator accumulates multiplicative. For instance, we analyze a setting where the reference mode $\pi_{\text{ref}}$ is a uniform distribution over the vocabulary space $V$. The importance weight in this setting is given by the following equation.

$$w(\mathbf{x}, \mathbf{y}) = |V|^T \prod_{t=1}^{T} \pi_\theta(\mathbf{y}_t|\mathbf{x}, \mathbf{y}_{<t})$$

The variance of the importance weights can grow exponentially large with respect to the number of tokens in the response $\mathbf{y}$.

$$\operatorname*{Var}_{\mathbf{y} \sim \pi_{\text{ref}}(\cdot|\mathbf{x})}[w(\mathbf{x}, \mathbf{y})] = |V|^{2T} \operatorname*{Var}_{\mathbf{y} \sim \pi_{\text{ref}}(\cdot|\mathbf{x})}\left[\prod_{t=1}^{T} \pi_\theta(\mathbf{y}_t|\mathbf{x}, \mathbf{y}_{<t})\right].$$

By using exponential smoothing importance weights

$$w(\mathbf{x}, \mathbf{y}) = |V| \prod_{t=1}^{T} \pi_\theta(\mathbf{y}_t|\mathbf{x}, \mathbf{y}_{<t})^\alpha$$

and choosing the value of $\alpha = \frac{1}{T}$, the variance of the importance weights is reduced significantly and does not grow exponentially with respect to the number of tokens in the response $y$.

$$\operatorname{Var}(w(x, y)) = |V|^2 \operatorname{Var}(\pi_\theta(y|x)^\alpha)$$

# F   ABALATION STUDY OF $\alpha$ VALUES

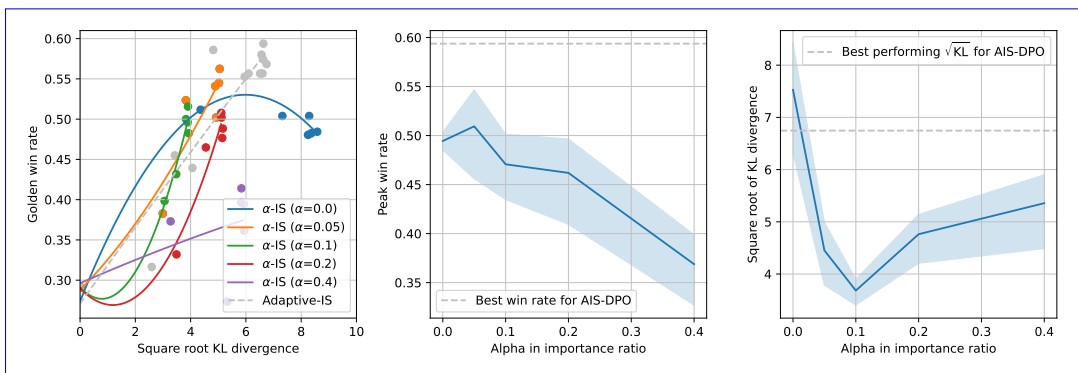

Figure 8: (Left) Win rate-KL tradeoff of different $\alpha$ values, we observed no over-optimization phenomenon and can even outperform DPO with the right $\alpha$, (Middle) the best win-rate of different $\alpha$ values, where $\alpha$ around 0.05 achieve the best performance. (Right) Best performing square root KL divergence of different $\alpha$ values. Increasing $\alpha$ helps regularization up to a specific point, the regularization effect will diminish when increasing $\alpha$ due to high variance issues.

**Fixed** $\alpha$.    We have provided an ablation over alpha in alignment experiments, we first fix $\beta = 0.01$ and vary $\alpha = (0.0, 0.05, 0.1, 0.2, 0.4)$ and compare with the adaptive-IS DPO and DPO objective on the Reddit TL;DR dataset. Figure 8 shows that a small value of $\alpha$ can achieve the best performance

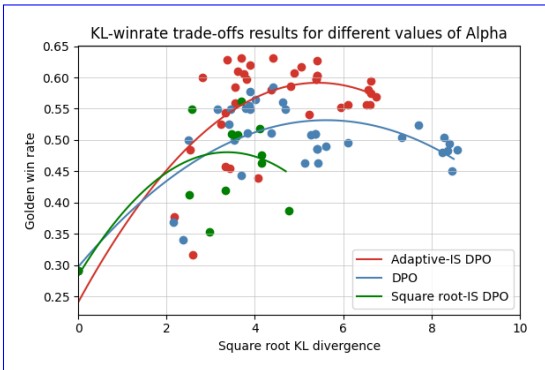

Figure 9: Win rate-KL tradeoff of DPO, AIS-DPO and different functional forms of $\alpha = \frac{1}{\sqrt{|y|}}$ (Square root-IDS DPO). Although Square-root IDS achieves a lower win rate than the other 2 methods, it still maintains a better regularization effect with the lowest KL budget.

with a lower KL budget than DPO. While increasing $\alpha$ helps increase the regularization effect and win rate. Up to a specific point (around 0.1), the regularization effect starts to diminish due to high variance in the importance ratio, causing unstable training.

**Adaptive** $\alpha$.    As mentioned in Section E, setting $\alpha = \frac{1}{|y|}$ can reduce variance of importance weighted estimators. We present the effect of this choice of $\alpha$ compared to fixed $\alpha$ values in Figure 8. Adaptive-IS achieves the best result in this setting while avoiding manually tunning the smoothing factor $\alpha$. We also provide experiment results for different functional forms of $\alpha$ that depend on the response length: $\alpha = \frac{1}{\sqrt{|y|}}$. Figure 9 shows that $\alpha = \frac{1}{\sqrt{|y|}}$ achieves a lower win rate than DPO and AIS-DPO. We speculate that setting $\alpha = \frac{1}{\sqrt{y}}$ can still have a high variance in the importance ratio, leading to a small number of samples having enormous weights that can potentially dominate learning signals of other valuable samples (Park et al., 2024b).

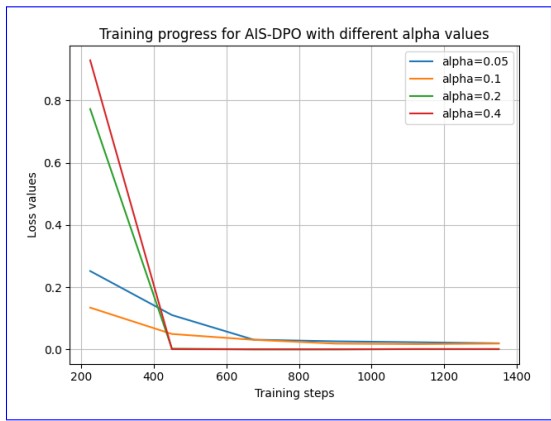

Figure 10: DPO loss over the course of training.

**Training convergence.** We plot the training loss of AIS-DPO with different $\alpha$ values during training in Figure 10 and observed that the training is stable in all settings. Another observation is that larger values of $\alpha$ lead to faster convergence than smaller values.

# G  HOW DOES ADAPTIVE IS PERFORM UNDER DISTRIBUTION SHIFT?

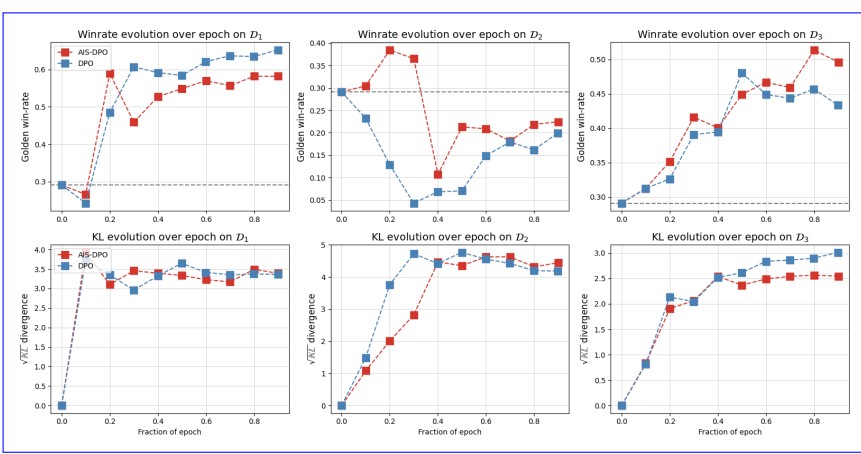

Figure 11: Win-rate and KL divergence against the fraction of epochs results for different shifted datasets $\mathcal{D}_1, \mathcal{D}_2, \mathcal{D}_3$ of Adaptive-DPO and standard DPO.

In this section, we provide an analysis of applying IS on the distribution shift experiment (presented in section 4.3). The results are plotted in Figure 11. When the data distribution is close to $\pi_{\text{ref}}$, Adaptive IS and DPO show similar performance in terms of win rate and KL divergence, but both still suffer from the distribution shift effect. Interestingly, we observed that as the data distribution shifts away from $\pi_{\text{ref}}$, AIS-DPO is shown to achieve better regularization and win rate compared to standard DPO even when the data is no longer from $\pi_{\text{ref}}$. This phenomenon is helpful in practice, where the preference data is usually generated from an unknown policy $\mu$, not from $\pi_{\text{ref}}$. AIS-DPO can still improve performance and regularization when $\pi_{\text{ref}}$ is not far from $\mu$.

# H  A POLICY GRADIENT DERIVATION OF DAAS

Our motivation to derive the original equation comes from the fact that DAAs method can be derived from vanilla policy gradient (VPG) ([4], [5]), an on-policy algorithm aims to maximize the following formula:

$$\nabla \mathcal{L}^{\mathrm{PG}}(\pi_\theta) = \mathbb{E}_{y \sim \pi_\theta(\cdot|x)} \left[ r(x,y) \nabla \log \pi_\theta(y|x) \right]$$

$$= \sum_y \nabla \pi_\theta(y|x) r(x,y)$$

The estimator above can have high variance, a popular approach is to subtract a baseline $b(x)$ to reduce variance while keeping the estimator unbiased. A popular choice of the baseline is $b(x) = \sum_y r(x,y) \pi_\theta(y|x)$. Plugging into the above equation, we get

$$\nabla \mathcal{L}^{\mathrm{PG}}(\pi_\theta) = \left( \sum_{y_1} r(x,y_1) \nabla \pi_\theta(y_1|x) - \sum_{y_1,y_2} r(x,y_2) \pi_\theta(y_2|x) \nabla \pi_\theta(y_1|x) \right)$$

$$= \sum_{y_1,y_2} r(x,y_1) \pi_\theta(y_2|x) \nabla \pi_\theta(y_1|x) - \sum_{y_1,y_2} r(x,y_2) \pi_\theta(y_2|x) \nabla \pi_\theta(y_1|x)$$

$$= \sum_{y_1,y_2} \left( r(x,y_1) - r(x,y_2) \pi_\theta(y_2|x) \nabla \pi_\theta(y_1|x) \right)$$

$$= \mathbb{E}_{(y_1,y_2) \sim \pi_\theta(\cdot|x)} \left[ (r(x,y_1) - r(x,y_2)) \nabla \log \pi_\theta(y_1|x) \right]$$

We then swapped actions $y_1, y_2$ and averaged them together to get the desired form.

$$\nabla \mathcal{L}^{\mathrm{PG}}(\pi_\theta) = \mathbb{E}_{(y_1,y_2) \sim \pi_\theta} \left[ \frac{(r(x,y_1) - r(x,y_2))}{2} \left( \nabla \log \pi_\theta(y_1|x) - \nabla \log \pi_\theta(y_2|x) \right) \right]$$

Where $r(x,y) = R(x,y) - \beta \log \frac{\pi_\theta(y|x)}{\pi_{\mathrm{ref}}(y|x)}$ is the reward-KL regularization in RLHF. The above equation is called *Pairwise Policy Gradient* and has been recently proposed by (Wu et al., 2024; Flet-Berliac et al., 2024).

Here we will show how Policy Gradient can be related to DAAs methods (e.g. IPO).

**Property** (IPO as Policy Gradient with binarized reward): Given a prompt $x$ and a pair of generations $(y_1, y_2)$, assuming that $y_1 \succ y_2$ and defining reward $R(x,y_1) = -R(x,y_2) = \frac{1}{4}$. Then we have**

$$\nabla \mathcal{L}^{\mathrm{VPG}}(\pi_\theta) = -\frac{1}{2\beta} \nabla \mathcal{L}_{\mathrm{IPO}}(\pi_\theta, \pi_{\mathrm{ref}})$$

*Proof*: The gradient of Policy Gradient can now be written:

$$\nabla \mathcal{L}^{\mathrm{PG}}(\pi_\theta) = \mathbb{E}_{(y_1,y_2) \sim \pi_\theta(\cdot|x)} \left[ \left( \frac{1}{2} - \beta \log \frac{\pi_\theta(y_1|x)}{\pi_{\mathrm{ref}}(y_1|x)} + \beta \log \frac{\pi_\theta(y_2|x)}{\pi_{\mathrm{ref}}(y_2|x)} \right) \left( \nabla \log \pi_\theta(y_1|x) - \nabla \log \pi_\theta(y_2|x) \right) \right]$$

Let's consider the gradient of IPO, a popular loss in DAAs family:

$$\nabla \mathbb{E}_{(y_1, y_2) \sim \pi_\theta(\cdot|x)} \left[ \left( \frac{1}{2} - \beta \left( \log \frac{\pi_\theta(y_1|x)}{\pi_{\text{ref}}(y_1|x)} - \log \frac{\pi_\theta(y_2|x)}{\pi_{\text{ref}}(y_2|x)} \right) \right)^2 \right]$$

$$= \mathbb{E}_{(y_1, y_2 \sim \pi_\theta(\cdot|x)} \left[ \nabla 2 \left( \frac{1}{2} - \beta \left( \log \frac{\pi_\theta(y_1|x)}{\pi_{\text{ref}}(y_1|x)} - \log \frac{\pi_\theta(y_2|x)}{\pi_{\text{ref}}(y_2|x)} \right) \right) (\nabla \log \pi_\theta(y_1|x) - \nabla \log \pi_\theta(y_2|x)) \right]$$

$$= -2\beta \mathcal{L}^{\text{PG}}(\pi_\theta)$$

As shown above, DAAs can be seen as maximizing binarized rewards with policy gradient. However, this equivalence only holds when we consider the online version of DPO or IPO. In off-policy setups, DAAs can suffer from the distribution shift problem, which has been well-studied in Offline RL literature Levine et al. (2020). This also explains the ineffectiveness of regularization in DAAs when using offline data due to the sampling bias in the regularization objective (Levine et al., 2020; Tang et al., 2024d).

Thus, Offline DAAs methods should be seen as on-policy maximizing expected reward $r(x, y)$ under the current LLM policy with the additional constraint that we only have access to some static dataset $\mathcal{D}$. Therefore, our ideal objective is:

$$\max_\theta J(\theta) = \mathbb{E}_{x \sim \mathcal{D}, (y_w, y_l) \sim \pi_\theta(\cdot|x)} \left[ \log \sigma \left( \beta \log \frac{\pi_\theta(y_w|x)}{\pi_{\text{ref}}(y_w|x)} - \beta \log \frac{\pi_\theta(y_l|x)}{\pi_{\text{ref}}(y_l|x)} \right) \right]$$

As we only have access to a static dataset generated from $\pi_{\text{ref}}$, we propose to use importance sampling to estimate expectations under $\pi_\theta$ distribution given samples from a reference distribution $\pi_{\text{ref}}$:

$$J(\theta) = \sum_{x, y_w, y} \pi_\theta(y_w|x) \pi_\theta(y_l|x) \left( \log \sigma \left( \beta \log \frac{\pi_\theta(y_w|x)}{\pi_{\text{ref}}(y_w|x)} - \beta \log \frac{\pi_\theta(y_l|x)}{\pi_{\text{ref}}(y_l|x)} \right) \right)$$

$$= \sum_{x, y_w, y} \pi_{\text{ref}}(y_w|x) \pi_{\text{ref}}(y_l|x) \frac{\pi_\theta(y_w|x)}{\pi_{\text{ref}}(y_w|x)} \frac{\pi_\theta(y_l|x)}{\pi_{\text{ref}}(y_l|x)} \left( \log \sigma \left( \beta \log \frac{\pi_\theta(y_w|x)}{\pi_{\text{ref}}(y_w|x)} - \beta \log \frac{\pi_\theta(y_l|x)}{\pi_{\text{ref}}(y_l|x)} \right) \right)$$

$$= \mathbb{E}_{(x, y_w, y_l) \sim \pi_{\text{ref}}} \left[ \frac{\pi_\theta(y_w|x)}{\pi_{\text{ref}}(y_w|x)} \frac{\pi_\theta(y_l|x)}{\pi_{\text{ref}}(y_l|x)} \log \sigma \left( \beta \log \frac{\pi_\theta(y_w|x)}{\pi_{\text{ref}}(y_w|x)} - \beta \log \frac{\pi_\theta(y_l|x)}{\pi_{\text{ref}}(y_l|x)} \right) \right]$$

Asuming that $\pi_\theta$ and $\pi_{\text{ref}}$ have the same support.

## I    RESULTS WITH REGULARIZED PREFERENCE OPTIMIZATION

Regularized Preference Optimization (RPO) (Liu et al., 2024) also shows that reward over-optimization happens due to distribution shift problem, similar to ours. They propose a theoretical algorithm that minimizes the DPO loss and an additional SFT term to mitigate reward over-optimization. The additional SFT loss ensures alignment with the reference policy to stabilize training and reduce uncertain labels in preference data.

On the other hand, we propose to mitigate distribution shift problem by adding an importance ratio to estimate samples under the current LM policy $\pi_\theta$. The importance ratio will upweight samples that have high likelihood under $\pi_\theta$ and downweight low likelihood samples. RPO also requires additional hyper-parameters $\eta$ to balance the tradeoff between alignment with the reference policy and learning from preference while our approach does not introduce any new hyper-parameters where the $\alpha$ terms adaptively trading off between bias

To compare Adaptive-IS with RPO, we use a similar experimental setup as described in section D. We train RPO with 3 different values of $\beta = (0.01, 0.05, 0.1)$ and tune $\eta = (0.001, 0.005, 0.01)$.

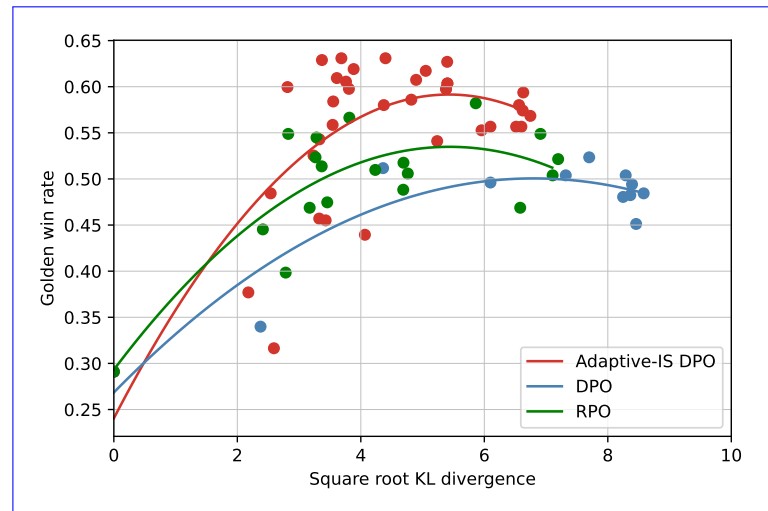

Figure 12: Win rate-KL tradeoff of Adaptive-IS, DPO, and RPO. Adaptive-IS achieves the best performance compared to other methods while maintaining a lower KL budget.

Based on the results, we select $\eta = 0.005$ as it yields the best performance and is consistent with the choice in the original study.

Figure 12 shows the win rate-KL tradeoff of Adaptive-IS, RPO, and DPO. As expected, DPO achieved the lowest performance with a higher KL divergence. Adaptive-IS DPO was able to show superior performance than RPO under a similar KL budget without requiring any additional hyper-parameters.

## J  DAAS REGULARIZATION SUFFERS FROM DISTRIBUTION SHIFT

In this section, we will follow Tang et al. (2024d) derivation to show how does DAAs methods suffer from distribution shift. As an example, given a prompt $x$. Let's consider the Taylor expansion of $\rho_\theta$ around zero in IPO loss:

$$\min_{\pi_\theta} \mathbb{E}_{(y_w, y_l) \sim \pi_{\text{ref}}(\cdot|x)} \left[ \left( \rho_\theta(y_w, y_l) - \frac{1}{2\beta} \right)^2 \right] = \frac{1}{4\beta^2} - \underbrace{\frac{1}{\beta} \mathbb{E}_{(y_w, y_l) \sim \pi_{\text{ref}}} \left[ \rho_\theta(y_w, y_l) \right]}_{\text{Preference Optimization}} + \underbrace{\mathbb{E}_{(y_w, y_l) \sim \pi_{\text{ref}}} \left[ \rho_\theta(y_w, y_l)^2 \right]}_{\mu - \text{weighted loss}}$$

The second term serves as regularization, which is called the $\mu$-weighted loss Tang et al. (2024d) ,this loss encourages $\pi_\theta$ to stay close to $\pi_{\text{ref}}$. Although, both KL divergence and $\mu$-weighted loss achieve the same global minimizer (when $\pi_\theta = \pi_{\text{ref}}$), their main difference lies in the their gradient. For KL divergence, $\nabla_\theta \text{KL}(\pi_\theta, \pi_{\text{ref}}) = \mathbb{E}_{y \sim \pi_\theta} \left[ \log \frac{\pi_\theta(y|x)}{\pi_{\text{ref}}(y|x)} \nabla \log \pi_\theta(y|x) \right]$, While the gradient of the $\mu$-squared loss:

$$\mathbb{E}_{(y_1, y_2) \sim \pi_{\text{ref}}(y|x)} \left[ \log \frac{\pi_\theta(y|x)}{\pi_{\text{ref}}(y|x)} \nabla \log \pi_\theta(y|x) \right]$$

There is a mismatch between data distribution, where KL divergence is calculated based on the current LM samples, while Offline regularization in DAAs directly use offline samples for regularization. This leads to cases minimize offline regularization objective might not necessarily minimize KL divergence. This is because the offline samples may not accurately represent the samples generated under the current policy, potentially causing performance degradation (Tang et al., 2024d; Chen et al., 2024a).

