# OpenReview forum: "Mitigating Reward Over-optimization in Direct Alignment Algorithms with Adaptive Importance Sampling"
_ICLR.cc/2025/Conference — Submitted to ICLR 2025_

### Official Review · Reviewer_WtSX · 2024-10-27

**Soundness:** 1
**Presentation:** 2
**Contribution:** 2
**Rating:** 3
**Confidence:** 4

**Summary:**

This paper studies the reward-optimization issue in aligning large language models. It focuses on direct alignment methods, such as Direct Preference Optimization (DPO) and Identity Preference Optimization (IPO). The paper argues that these issues arise from off-policy distribution shifts between the learning policy and the reference policy. Accordingly, an importance-sampling weighting term with adaptive schemes is proposed. Experiments with Pythia models on the TL;DR dataset are conducted.

**Strengths:**

- The idea of using importance sampling to address distribution shift is not new, but it sounds interesting in the context of direct alignment methods.
- This paper is well-written and easy to follow.
- Numerous empirical results are presented, along with their limitations (see below).

**Weaknesses:**

- This paper lacks technical depth. It studies the distribution shift issue in DPO, which is a valuable perspective. Unfortunately, it fails to explicitly point out or mention that DPO's gradient estimator is not unbiased because the data distribution is defined by the data-collection distribution policy $\pi$ (see previous works [1, 2]). Furthermore, it fails to justify that the proposed gradient estimator is unbiased. The reviewer believes that it is not theoretically unbiased. In fact, the importance sampling weight requires the optimal policy $\pi^*$, which is not available a priori.

[1] Liu, Tianqi, et al. "Statistical rejection sampling improves preference optimization." *arXiv preprint arXiv:2309.06657* (2023).

[2] Xiong, Wei, et al. "Iterative preference learning from human feedback: Bridging theory and practice for RLHF under KL-constraint." *Forty-first International Conference on Machine Learning*. 2024.

From the reviewer's perspective, there are two factors in DPO's formulation that prevent it from finding the true optimal policy:

First, DPO uses a fixed and offline dataset, where the data distribution does not originate from the optimal policy.

Second, DPO employs KL regularization with a fixed policy. To address these issues, two simple strategies can be applied: periodically updating the reference policy [3] or using entropy regularization [4].

[3] Guo, Shangmin, et al. "Direct language model alignment from online AI feedback." *arXiv preprint arXiv:2402.04792* (2024).

[4] Xiao, Jiancong, et al. "On the Algorithmic Bias of Aligning Large Language Models with RLHF: Preference Collapse and Matching Regularization." *arXiv preprint arXiv:2405.16455* (2024).

- The superiority over other simple baselines is unclear. A straightforward way to address the distribution shift issue is to use a moving average of the reference policy that can ensure the policy moving beyond the KL contraint.

- Experimental results are weak. The experiments are conducted on the TL;DR dataset, which unfortunately has very short response lengths, and the Pythia model used as a base is quite weak. Consequently, empirical conclusions and insights may have limited value for modern language models. Moreover, some experiment details are missing, which hinders reproducibility and understanding of key results.

**Questions:**

1. Can the authors theoretically justify that the estimator is unbiased?

2. Can the authors discuss the issue of length bias in the importance sampling weight? The length bias affects reward estimation [5], so the reviewer wonders about its effect on the importance sampling estimator used in this paper.

[5] Park, Ryan, et al. "Disentangling length from quality in direct preference optimization." arXiv preprint arXiv:2403.19159 (2024).

3. Can this paper provide comparisons with online algorithms such as PPO, REINFORCE, and online DPO? Except for PPO, which requires extensive computational resources, other methods require nearly the same resources as DPO.

4. Can the authors clarify the methods used for KL calculation and "gold win-rate" calculation in the experiments?

---

> ### Author Response · Authors · 2024-11-22
>
> Thank you for your thoughtful review and valuable feedback. Please find our response to your comments below.
>
> **1. Can the authors theoretically justify that the estimator is unbiased? How does importance sampling estimator depend on the optimal policy $\pi^{*}$?**
>
> To show why importance sampling is unbiased. From [10], we have the objective
> $$
> \max\_{\pi^*} \mathbb{E}\_{x\sim \mathcal D, (y\_w,y\_l)\sim\pi^*(\cdot|x)}\left[\log\sigma\left(\beta\log\frac{\pi^*(y\_w|x)}{\pi\_{\text{ref}}(y\_w|x)}-\beta\log\frac{\pi^*(y\_l|x)}{\pi\_{\text{ref}}(y\_l|x)}\right)\right]
> $$
>
> In RSO, they propose to use rejection sampling to approximate the optimal policy in order to construct preference data $\mathcal D\_p=\{(x,y\_w,y\_l)|y\_w,y\_l\sim\pi^*(\cdot|x)\}$ and then fitting a parametrized policy $\pi\_\theta$ with maximum likelihood objective, resulting in:
>
> $$
> \max\_{\pi\_\theta}\mathbb{E}\_{x\sim \mathcal P, (y\_w,y\_l)\sim\pi^*(\cdot|x)}\left[\log\sigma\left(\beta\log\frac{\pi\_\theta(y\_w|x)}{\pi\_{\text{ref}}(y\_w|x)}-\beta\log\frac{\pi\_\theta(y\_l|x)}{\pi\_{\text{ref}}(y\_l|x)}\right)\right]
> $$
>
> Instead of using RSO to sample from the optimal policy, one can directly parameterize the optimal policy as $\pi\_\theta$ for collecting preference data and reward learning (as shown in [10, 12]), resulting in:
>
> $$
> \max\_{\pi\_\theta} J(\theta) = \mathbb{E}\_{x\sim \mathcal P, (y\_w, y\_l)\sim\pi\_\theta(\cdot|x)}\left[\log\sigma\left(\beta\log\frac{\pi\_\theta(y\_w|x)}{\pi\_{\text{ref}}(y\_w|x)}-\beta\log\frac{\pi\_\theta(y\_l|x)}{\pi\_{\text{ref}}(y\_l|x)}\right)\right]
> $$
>
> This corresponding to Online-DPO objective, where the parametrized policy $\pi\_\theta$ is now serve as an approximation to $\pi^*$ for generating preference data and reward learning. This is also our desire objective.
>
> Asuming that $\pi\_\theta$ is fully supported on $\pi\_{\text{ref}}$, the online-DPO objective can be written:
>
> $$
> \begin{align*}
> J(\theta)&=\sum\_{x,y\_w,y}\pi\_\theta(y\_w|x)\pi\_\theta(y\_l|x)\left(\log\sigma\left(\beta\log\frac{\pi\_\theta(y\_w|x)}{\pi\_{\text{ref}}(y\_w|x)}-\beta\log\frac{\pi\_\theta(y\_l|x)}{\pi\_{\text{ref}}(y\_l|x)}\right)\right)\\\\
> &=\sum\_{x,y\_w,y}\pi\_{\text{ref}}(y\_w|x)\pi\_{\text{ref}}(y\_l|x)\frac{\pi\_\theta(y\_w|x)}{\pi\_{\text{ref}}(y\_w|x)}\frac{\pi\_\theta(y\_l|x)}{\pi\_{\text{ref}}(y\_l|x)}\left(\log\sigma\left(\beta\log\frac{\pi\_\theta(y\_w|x)}{\pi\_{\text{ref}}(y\_w|x)}-\beta\log\frac{\pi\_\theta(y\_l|x)}{\pi\_{\text{ref}}(y\_l|x)}\right)\right)\\\\
> &=\mathbb E\_{(x,y\_w,y\_l)\sim\pi\_{\text{ref}}}\left[\frac{\pi\_\theta(y\_w|x)}{\pi\_{\text{ref}}(y\_w|x)}\frac{\pi\_\theta(y\_l|x)}{\pi\_{\text{ref}}(y\_l|x)}\log\sigma\left(\beta\log\frac{\pi\_\theta(y\_w|x)}{\pi\_{\text{ref}}(y\_w|x)}-\beta\log\frac{\pi\_\theta(y\_l|x)}{\pi\_{\text{ref}}(y\_l|x)}\right)\right]
> \end{align*}
> $$
>
> which shows that DPO with Importance sampling (our objective) is an unbiased estimator of Online-DPO.
>
>
> **2. Can this paper provide comparisons with online algorithms such as PPO, REINFORCE, and online DPO? Except for PPO, which requires extensive computational resources, other methods require nearly the same resources as DPO.**
>
> We conducted further experiments where we compare DAAs with online alignment methods. As PPO requries extensive computational resources, we consider an alternatives online algorithms which is REINFORCE Leave-One-Out (RLOO) [7]. Detail training setting and a figure presenting the result can be found in our updated manuscript (Section C in the Appendix). As expected, RLOO achives better win-rate compared to DPO and AIS-DPO and utilizing better KL budget. The result also shows that AIS-DPO help close the gap between offline and online algorithms.

---

> ### Author Response · Authors · 2024-11-22
>
> **3. Can the authors clarify the methods used for KL calculation and "gold win-rate" calculation in the experiments?**
>
> The calculation of KL divergence in our paper is based on [8] and the KL is estimated by taking on-policy samples from the current LM $\pi\_\theta$.
>
> Specifically, we first sample $N$ input prompts $\\{\mathbf{x}\_i\\}\_{i = 1}^N$ from the evaluation set. For each input prompt $\mathbf{x}\_i$, we generate a response $\mathbf{y}\_i$ using the current policy $\pi\_\theta$. Let $T\_i$ be the length of the response $\mathbf{y}\_i$, we compute the KL divergence between $\pi\_\theta$ and $\pi\_{\text{ref}}$ as follows:
>
> $$
> \begin{align*}
> \frac{1}{N}\sum\limits\_{n=1}^N\sum\limits\_{t=1}^T\mathbb{KL}\big(\pi\_\theta(\cdot|\mathbf{x,y}\_{<t}), \pi\_{\text{ref}}(\cdot|\mathbf{x,y}\_{<t})\Big)
> \end{align*}
> $$
>
> We set $N=512$ in our experiments.
>
> For Gold win-rate calculation, we first use a well fine-tuned pythia 6.9b in [9]. The model achieve $\approx 70$% accuracy in evaluation set and achieving 76.7% agreement ratio with ChatGPT Evaluation using LLM as a judges prompting.
>
> For a given prompt $x$, we first sample a response $y\sim\pi\_\theta(\cdot|x)$ and then use the golden reward model $r^{\text{gold}}$  to compare against reference summaries $y\_{\text{ref}}$ in evaluation set to determine the win-rate with the following calculation:
>
> $$
> \frac{1}{N}\sum\limits\_{i=1}^N \boldsymbol{1}\\{r^{\text{gold}}(x,y) > r^{\text{gold}}(x,y\_{\text{ref}})\\}
> $$
>
> We have included the calculation of KL divergence and gold win-rate calculation in Section B, Appendix in the revised manuscript.
>
> **4. Can the authors discuss the issue of length bias in the importance sampling weight? The length bias affects reward estimation [5], so the reviewer wonders about its effect on the importance sampling estimator used in this paper.**
>
> The length of responses could affect the variance of the importance sampling estimators. Since our importance weights are computed as the ratio of probabilities from two distributions, accumulated over all tokens. The variance of importance sampling estimators can grow exponentially with the number of tokens in the output. To address this, we applied exponential smoothing of importance weights, using a smoothing factor $\alpha = \frac{1}{T}$. This adjustment reduces the variance significantly, preventing exponential growth with the response length. We provide a detailed analysis in the general response and in Sections E and F of the Appendix in our revised manuscript.
>
> **5. The superiority over other simple baselines is unclear. A straightforward way to address the distribution shift issue is to use a moving average of the reference policy that can ensure the policy moving beyond the KL contraint.**
>
> While it is straightforward to address the distribution shift issue is to periodically updating the reference policy [5] or using entropy regularization [6]. Both of these methods require online-sample from the current LM policy in the loop of training and and explicit reward model, incurring significant computational costs.
>
> While in our paper, we working on an offline setting, we only have access to a static offline dataset. we propose to use importance sampling to mitigate distribution shift problem by adding a simple weighting to simulate Online RLHF without requiring generating samples and explicit reward model, which is computationally expensive.

---

> > ### Author Response · Authors · 2024-11-22
> >
> > **6. Experimental results are weak. The experiments are conducted on the TL;DR dataset, which unfortunately has very short response lengths, and the Pythia model used as a base is quite weak. Consequently, empirical conclusions and insights may have limited value for modern language models. Moreover, some experiment details are missing, which hinders reproducibility and understanding of key results.**
> >
> > Due to resources limitation, we cannot experiments for bigger model. At submisison time, strong models such as Llama-3.2 has not release, so we cannot experiments on  these stronger model. We follow closely experiments setting of ([1], [2], [3]), which use Pythia models and TL;DR for reward over-optimization experiments. We will plan to extend this in the future using HH-RLHF dataset and using strong models such as Llama-3.2 models family.
> >
> > We have provided experiments details of traning process, data pre-processing, details calculation of KL divergence and gold win-rate calculation in the revised manuscript.
> >
> > ----
> >     [1] Scaling Laws for Reward Model Overoptimization in Direct Alignment Algorithms
> >     [2] Disentangling Length from Quality in Direct Preference Optimization
> >     [3] Understanding Preference Fine-Tuning Through the Lens of Coverage
> >     [5] Guo, Shangmin, et al. "Direct language model alignment from online AI feedback." arXiv preprint arXiv:2402.04792 (2024).
> >     [6] Xiao, Jiancong, et al. "On the Algorithmic Bias of Aligning Large Language Models with RLHF: Preference Collapse and Matching Regularization." arXiv preprint arXiv:2405.16455 (2024).
> >     [7] Back to Basics: Revisiting REINFORCE Style Optimization for Learning from Human Feedback in LLMs
> >     [8] Understanding the performance gap between online and offline alignment algorithms.
> >     [9] The N+ Implementation Details of RLHF with PPO: A Case Study on TL;DR Summarization.
> >     [10] SAIL: Self-Improving Efficient Online Alignment of Large Language Models
> >     [11] Statistical rejection sampling improves preference optimization.
> >     [12] Direct Preference Optimization: Your Language Model is Secretly a Reward Model
> > ----
> >
> > We hope we have cleared your concerns about our work. We have also revised our manuscript according to your comments, and we would appreciate it if we can get your further feedback.

---

> > > ### Comment · Reviewer_WtSX · 2024-11-25
> > > **Thanks for clarification**
> > >
> > > Thank you for your detailed response. I have checked the revised paper and I appreciate the clarification. However, my concerns are not addressed.
> > >
> > > A minor comment in the mathmatical derivation: it misses a validation: the assumption that the preference randomness over $y_w$ and $y_l$ is independent of $\pi_{\text{ref}}$.
> > >
> > > Additionally, while the paper claims that this estimator is unbiased for online DPO, my concern pertains to the unbiasedness of the true gradient when sampling from the optimal policy $\pi^{*}$. A critical point is that the dataset must provide sufficient coverage of $y_w$ and $y_l$, but the importance sampling technique does not enhance this coverage.
> > >
> > > Furthermore, it should be clarified that IS-DPO approximates only a single step of online DPO, whereas the full online DPO framework involves multiple iterations to refine approximations of $\pi^{*}$. Importantly, the role of importance sampling is not central within the online DPO framework. As such, my concerns remain: importance sampling is insufficient to address the fundamental challenges of offline learning.
> > >
> > > Finally, I cannot see the potential value of this method in addressing the needs of modern language models in the future. With larger models and longer sequence lengths, the variance issues inherent in importance sampling become increasingly problematic. To address this, the authors incorporate Exponential Smoothing Importance Sampling from prior work (Aouali et al., 2023; Korba & Portier, 2022) as a mitigation strategy. However, this introduces additional bias in gradient estimation and further complicates the approach by requiring hyperparameter tuning. Moreover, as the paper primarily relies on the direct application of existing techniques, it appears to fall short of the standards typically expected for publication at ICLR.
> > >
> > > Given these concerns, I find it difficult to identify any significant technical or practical impact of this method and have decided to maintain my recommendation.

---

> > > > ### Author Response · Authors · 2024-11-27
> > > >
> > > > We thank the reviewer for their helpful feedback and will respond to the raised questions below.
> > > >
> > > > **1. A minor comment in the mathematical derivation: it misses a validation: the assumption that the preference randomness over $y_w$ and $y_l$  is independent of $\pi_{\text{ref}}$.**
> > > >
> > > > We appreciate the reviewer for the comments about the mathematical derivation. We will revise this part accordingly in our final version of the paper.
> > > >
> > > > **2. Additionally, while the paper claims that this estimator is unbiased for online DPO, my concern pertains to the unbiasedness of the true gradient when sampling from the optimal policy $\pi^\*$. A critical point is that the dataset must provide sufficient coverage of $y_w$ and $y_l$, but the importance sampling technique does not enhance this coverage.**
> > > >
> > > > We completely agree with the reviewer that one issue of offline DAAs is the data coverge problem. It's extremely challenging for the offline dataset to sufficiently cover the support of the optimal policy. There are methods such as Statistical Rejection Sampling Optimization (RSO) which addresses the coverage problem by applying rejection sampling using SFT policy and a trained reward model [2].
> > > >
> > > > However, as empirically shown in a recent study (Hypothesis 1 in [1]), **the performance gap between online and offline DAAs cannot be explained by the difference in data coverage alone**. Specifically, assuming that the data-collection policy and the optimal policy are different but share the same support, thus the offline data can provide sufficient coverage. The analysis in Section 2.4 and Section 3.2 suggests that during training, there is still a mismatch between the KL regularization in online and offline alignment methods due to distribution shift, leading to the over-optimization phenomenon.
> > > >
> > > > In this work, we address the issue by using importance sampling to estimate the online alignment objectives using offline data sampled from a reference policy. Under the assumption that $\pi_\theta$ is fully supported on $\pi_\text{ref}$, the proposed IS estimator is unbiased, given by the following derivation
> > > >
> > > > $$
> > > > \begin{align*}
> > > > &\underbrace{\mathbb E\_{(y\_w,y\_l)\sim\pi\_{\text{ref}}}\left[\frac{\pi\_\theta(y\_w|x)}{\pi\_{\text{ref}}(y\_w|x)}\frac{\pi\_\theta(y\_l|x)}{\pi\_{\text{ref}}(y\_l|x)}\log\sigma\left(\beta\log\frac{\pi\_\theta(y\_w|x)}{\pi\_{\text{ref}}(y\_w|x)}-\beta\log\frac{\pi\_\theta(y\_l|x)}{\pi\_{\text{ref}}(y\_l|x)}\right)\right]}\_{\text{IS-DPO objective}} \\\\
> > > > =&\sum\_{y\_w,y\_l} \pi\_{\text{ref}}(y\_w|x)\pi\_{\text{ref}}(y\_l|x) \frac{\pi\_\theta(y\_w|x)}{\pi\_{\text{ref}}(y\_w|x)}\frac{\pi\_\theta(y\_l|x)}{\pi\_{\text{ref}}(y\_l|x)}\log\sigma\left(\beta\log\frac{\pi\_\theta(y\_w|x)}{\pi\_{\text{ref}}(y\_w|x)}-\beta\log\frac{\pi\_\theta(y\_l|x)}{\pi\_{\text{ref}}(y\_l|x)}\right) \\\\
> > > > =&\sum\_{y\_w,y\_l} \pi\_\theta(y\_w|x)\pi\_\theta(y\_l|x) \log\sigma\left(\beta\log\frac{\pi\_\theta(y\_w|x)}{\pi\_{\text{ref}}(y\_w|x)}-\beta\log\frac{\pi\_\theta(y\_l|x)}{\pi\_{\text{ref}}(y\_l|x)}\right) \\\\
> > > > =&  \underbrace{E\_{(y\_w,y\_l)\sim\pi\_\theta}\left[\log\sigma\left(\beta\log\frac{\pi\_\theta(y\_w|x)}{\pi\_{\text{ref}}(y\_w|x)}-\beta\log\frac{\pi\_\theta(y\_l|x)}{\pi\_{\text{ref}}(y\_l|x)}\right)\right]}\_{\text{Online DPO objective}} \text{ (1)}
> > > > \end{align*}
> > > > $$
> > > >
> > > > Our approach is orthogonal to the rejection sampling method proposed in [2] and can be combined with it to address the challenges of offline alignment. In our revised manuscript, we will include additional discussion on the issue of data coverage.
> > > >
> > > > **3. Furthermore, it should be clarified that IS-DPO approximates only a single step of online DPO, whereas the full online DPO framework involves multiple iterations to refine approximations of $\pi^\*$. Importantly, the role of importance sampling is not central within the online DPO framework. As such, my concerns remain: importance sampling is insufficient to address the fundamental challenges of offline learning.**
> > > >
> > > > From (1), we observe that, under the assumption of full support, the optimal policy for the IS-DPO objective is also optimal for the online DPO objective. In practice, online DPO is implemented as batch online DPO, which involves multiple iterations [4]. Each iteration samples a batch of data from the current policy, labels this batch, and then updates the current policy using the DPO objective with the new batch. Since $\pi_\text{ref}$ remains fixed in each iteration (as described in Algorithm 1 of [4]), the resulting policy from this process should converge to the optimal policy of the online DPO objective. Therefore, IS-DPO does not merely approximate a single step of online DPO.

---

> > > > > ### Author Response · Authors · 2024-11-27
> > > > >
> > > > > **4. Finally, I cannot see the potential value of this method in addressing the needs of modern language models in the future. With larger models and longer sequence lengths, the variance issues inherent in importance sampling become increasingly problematic.**
> > > > >
> > > > > We agree that for long sequences, the variance in IS can lead to a performance degrade, which we already demonstrated in our analysis and ablation studies in Sections E and F of the Appendix. To address this issue, we propose Adaptive-IS, which selects the smoothing factor $\alpha$ as the inverse of the sequence length, which helps reduce the variance in IS (please refer to our analysis in the General Response and in Section... of the updated paper). This approach eliminates the need for manual tuning of $\alpha$ while still achieving a favorable trade-off between Win rate and KL divergence. As a result, we believe that our method provides great value for training modern language models with DAAs.
> > > > >
> > > > > **In summary,** we have provided the reviewer with our explanation for the significant technical contributions of our work. Our method is also practical in solving the over-optimization phenomenon in DAAs in training LLMs, including the modern language models. We hope that these points have addressed the reviewer's concerns about our work. Nevertheless, we're happy to promptly any additional questions from the reviewer during the rebuttal period.
> > > > >
> > > > > ----
> > > > >     [1] Understanding the performance gap between online and offline alignment algorithms.
> > > > >     [2] Statistical rejection sampling improves preference optimization.
> > > > >     [3] Scaling Laws for Reward Model Overoptimization in Direct Alignment Algorithms
> > > > >     [4] Direct Language Model Alignment from Online AI Feedback
> > > > > ----

---

> > > > > > ### Author Response · Authors · 2024-12-01
> > > > > >
> > > > > > Dear Reviewer WtSX,
> > > > > >
> > > > > > As the rebuttal period is **coming to an end very shortly**, we again thank the Reviewer for the valuable initial comments and the additional questions, which we had previously responded during the rebuttal phase; all the corresponding changes are now incorporated in the revised paper during rebuttal.
> > > > > >
> > > > > > While we understand the Reviewer's point of view, we have also explained that our proposed algorithm, albeit being simple, is novel with rigorous theoretical foundations and empirical demonstrations provided by our work. We believe that this algorithm not only can help alleviate the overoptimization problem in any existing DAAs, but also can be easily integrated into any of these algorithms, further showing its practicality in real-world training using DAAs.
> > > > > >
> > > > > > We hope that the Reviewer can take into account the significance and many contributions of our work, along with our extensive efforts in the rebuttal phase, in the final rating. We sincerely hope the Reviewer will raise the score accordingly.
> > > > > >
> > > > > > Sincerely,
> > > > > > Paper 14149’s Authors

---

> > > > > > > ### Comment · Reviewer_WtSX · 2024-12-02
> > > > > > > **Thanks for Detailed Response**
> > > > > > >
> > > > > > > Thank you for your explanation. I have checked Appendices D and E for the analysis, but I find the arguments unconvincing due to their reliance on unrealistic assumptions. For instance, the analysis of the importance sampling ratio overlooks critical components, such as the random variable $\log \pi_{\theta}$ and related terms. Additionally, when discussing the exponential growth of variance in the special case (i.e., a uniform distribution for $\pi_{\operatorname{ref}}$), it seems the argument omits the condition that the probability of each response is exponentially small. As a result, these points lack rigor and fail to persuade me.
> > > > > > >
> > > > > > > Unforunately, the above explanation does not address my concerns. I still believe that strong numerical evidence on large models is necessary to justify the technique or, alternatively, a rigorous theoretical analysis should be provided as a valuable contribution. At present, the standard importance sampling technique with a length normalization trick does not appear to be a significant technical contribution for me.

---

> > > > > > > > ### Author Response · Authors · 2024-12-04
> > > > > > > > **Thanks for your further feedback**
> > > > > > > >
> > > > > > > > Thanks for your further feedback. Please allow us to address your concerns below.
> > > > > > > >
> > > > > > > > **Q1: I have checked Appendices D and E for the analysis, but I find the arguments unconvincing due to their reliance on unrealistic assumptions.**
> > > > > > > >
> > > > > > > > We'd like to explain that the statement "the assumptions made by our analysis in Sections D and E are unrealistic" **is not a correct description of the analysis in Section D and E**. More specifically, we made 2 assumptions in the analysis in Sections D and E: (1) $\pi_\theta$ and $\pi_{\text{ref}}$ have overlapping supports and (2) $\pi_{\text{ref}}$ is uniform distribution over the vocabulary space $V$.
> > > > > > > > The first assumption (1) (*overlapping support*) is also made in several works, including IPO [5] and DPO [6]. The second assumption (2) (*uniform $\pi_{\text{ref}}$*) denotes the worst case scenario where the variance grows exponential in the response length, which is made to demonstrate that even in the worst case, we can stills ensure the practicality of our proposed exponential smoothing; note that this worst-case assumption is also made in other works, such as [2] (ICML'16) and [7]. For these reasons, we believe that these assumptions are realistic and allow us to demonstrate the practicality of the proposed adaptive importance sampling.

---

> > > > > > > > > ### Author Response · Authors · 2024-12-04
> > > > > > > > >
> > > > > > > > > **Q2: The analysis of the importance sampling ratio overlooks critical components, such as the random variable $\log{\pi_\theta}$ and related terms. When discussing the exponential growth of variance in the special case (i.e., a uniform distribution for $\pi_{\text{ref}}$, it seems the argument omits the condition that the probability of each response is exponentially small.**
> > > > > > > > >
> > > > > > > > > First, we'd like to explain that our analysis does not overlook any critical components.
> > > > > > > > >
> > > > > > > > > In Sections D and E, we shows that the variance of $w(\mathbf{x},\mathbf{y})$ depends on the product of $|V|^{2T}$ and $\underset{{y \sim \pi_{\text{ref}}(\cdot|x)}}{\operatorname{Var}}\left[\prod_{t=1}^T\pi_\theta(y_t|x,y_{<t})\right]$, which "grows exponentially large w.r.t the number of tokens in the response ${\bf y}$". The proposed exponential smoothing of importance weight, $w(x,y) = |V| \prod_{t=1}^T\pi_\theta(y_t|x,y_{<t})^\alpha$, reduces this exponentially growth of the variance of the importance weight significantly because:
> > > > > > > > >
> > > > > > > > > * for the first term, it helps reduce $|V|^{2T}$ to $|V|^2$
> > > > > > > > > * for the second term ${\operatorname{Var}}\left[\prod_{t=1}^T\pi_\theta(y_t|x,y_{<t})\right]$, the variance when apply smoothing ${\operatorname{Var}}\left[\prod_{t=1}^T\pi_\theta(y_t|x,y_{<t})^\alpha\right]$ will also be reduced, given that $\alpha<1$. (as we have shown in our analysis in Section A).
> > > > > > > > >
> > > > > > > > > As a result, **our analysis actually didn't overlook any critical components such as the second term that involves $\pi_\theta$** as the Reviewer mentioned. We will make this part clearer in the final version of our paper.
> > > > > > > > >
> > > > > > > > > Second, we'd like to response to the second point made by the Reviewer: "the argument omits the condition that the probability of each response is exponentially small". The variance of the second term, i.e., $\text{Var}(\pi_\theta(y|x))$, can still be high due to the multiplicatively accumulation over many timesteps even though the probability of each response is exponentially small (please see the analysis in [3,4]). This can cause drastic changes or large updates to the policy $\pi_\theta$ from the reference model. More specifically, due to high variance issues of $(\pi_\theta(y|x))$, there still exist examples that have enormously large weights ($\approx 3e5$). This can causes unstable training, where the models is taking a really large update and can potentially dominate learning signals of other valuable samples. Consequently, this motivates us to address or correct this high variance in the design of the proposed smoothing term, Adaptive IS, which effectively mitigates this isuse.
> > > > > > > > >
> > > > > > > > > Finally, besides our theoretical analysis, we'd like to re-affirm the Reviewer about the role of the smoothing factor with empirical evidence. Specifically, We observe that the importance ratio indeed goes to zero as training progress, as shown in the Table below.
> > > > > > > > >
> > > > > > > > >
> > > > > > > > > |  | Step-20096| Step-40192 | Step-60288| Step-80384| Step-92800|
> > > > > > > > > | -------- | -------- | -------- | -------- | -------- | --------|
> > > > > > > > > | Importance ratio (averaged over a training batch)     | 0.54039     | 0.45813  | 0.3641 | 0.38989 | 0.24347
> > > > > > > > >
> > > > > > > > > The above table presents values of importance ratio when applying adaptive importance sampling with  $\alpha=\frac{1}{|y|}$. We observed that adaptive importance sampling helps stabilize training process, effectively solving high variance issues in standard importance sampling.
> > > > > > > > >
> > > > > > > > >
> > > > > > > > > ----
> > > > > > > > >     [1] Notes on Importance Sampling and Policy Gradient
> > > > > > > > >     [2] Jiang et al. Doubly Robust Off-policy Value Evaluation for Reinforcement Learning. ICML'16
> > > > > > > > >     [3] Levine et al. Offline Reinforcement Learning: Tutorial, Review, and Perspectives on Open Problems
> > > > > > > > >     [4] Precup et al. Eligibility Traces for Off-Policy Policy Evaluation
> > > > > > > > >     [5] Azar et al. A General Theoretical Paradigm to Understand Learning from Human Preferences. AISTATS'24
> > > > > > > > >     [6] Rafailov et al. Direct Preference Optimization: Your Language Model is Secretly a Reward Model
> > > > > > > > >
> > > > > > > > > ----

---

> ### Author Response · Authors · 2024-12-04
>
> **Q3: Unfortunately, (1) the above explanation does not address my concerns. (2) I still believe that strong numerical evidence on large models is necessary to justify the technique or, alternatively, (3) a rigorous theoretical analysis should be provided as a valuable contribution. At present, (4) the standard importance sampling technique with a length normalization trick does not appear to be a significant technical contribution for me.**
>
> We annotate the Reviewer's points ((1), (2), (3), and (4)) for easier references.
>
> As we extensively explained in our rebuttal (an incorporated in our revised paper), along with the response in **Q2** of this response, the design of our Adaptive IS is rigorously analyzed, both theoretically and empirically, which we hope it could resolve concerns (1), (3), and (4) of the Reviewer. Additionally, AIS addresses the reward over-optimization in existing DAAs, where the weak regualrization in DAAs stemming from the mismatch between training data distribution and the current LM policy, which is a significant technical contribution in this domain (as also appreciated by Reviewers u6Dx  and zEKh); we hope that for this reason, we answer the Reviewer's concern in (4).
>
> Finally, we'd like to defend our paper, focusing on the Reviewer's comments:
>
> * We closely follow the exact same experimental setup (including TL;DR dataset and the Pythia model) in many directly related works [1,3,11,5] that study reward over-optimization in RLHF and DAAs. The fact that AIS improves the performance of existing DAAs in this well-established experimental setup already demonstrates the significance of our contributions and the practicality of the proposed method.
>      * While we understand the Reviewer's suggestion to evaluate "on large models", we sincerely think that our contributions are significant and our method is practical, without such evaluation, similar to the acceptance of the related works [1,3,11,5]. Additionally, evaluating on larger models requires high computational resource and laborous effort as we also need to replicate many existing baselines on these large models. Nevertheless, we followed the Reviewer's suggestion on larger models. In the limited Rebuttal window, we evaluate our method with Pythia-2.8b and present the win-rate evolution as training progress in the below tables.
>
>     | Method | Step-14400 | Step-28800 |Step-43200| Step-57600| Step-72000| Step-92820 |
>     | -------- | -------- | -------- | -------- | -------- | -------- | -------- |
>     | DPO     |  0.658   | 0.753     | 0.472 | 0.371 | 0.564 | 0.675|
>     |Adaptive-IS DPO | 0.625 | 0.673 | 0.763 | 0.703 | 0.718 | 0.75|
>
>     The table below shows the KL divergence evolution as training progress with Pythia-2.8b model.
>
>     | Method | Step-14400 | Step-28800 |Step-43200| Step-57600| Step-72000| Step-92820 |
>     | -------- | -------- | -------- | -------- | -------- | -------- |-------- |
>     | DPO     |  9.92    | 15.673    | 26.184 | 32.59| 28.25 | 26.15 |
>     |Adaptive-IS DPO | 9.158 | 12.662 | 16.595 | 22.193 | 24.1 | 22.463 |
>
>     In this setting, Adaptive-IS still shows consistent performance as the training progress while DPO performance start to degrade. Moreover, Adaptive-IS also maintains better KL regularization.

---

> > ### Author Response · Authors · 2024-12-04
> >
> > * We's also like to note that the variance of the importance weights is accumulated with respect to the **response length**, not to the **whole sequence length** (which includes the prompt and response lenghts). As we increase the sequence length, it does not necessarily contribute to variance issues inherent in importance sampling.
> >       * In terms of response length, most studied alignment datasets including HH-RLHF [1], Reddit TL;DR [8] dataset, Chat arena sxs [4], WebGPT [9], AlpacaEval [10] primarily include response length with less than 200 tokens. These datasets has been used extensively for Evaluting Alignment in LLMs ([4], [1], [2], [3], [6], [5], [11]). Our work closely aligns with these benchmarks, ensuring both (i) coverage of a significant portion of current practical applications, and (ii) consequently the practicality of our proposed method.
> >       * Finally, when the response $\mathbf{y}$ is long, our adaptive smoothing factor will be small (typically $\le0.05$ for the Reddit TLDR dataset). Indeed, in our ablation study (Section F), we observed that small values of $\alpha$ achieve better regularization and performance.
> >
> > * We also followed the Reviewer's suggestion on "stronger" models (suggested in the original response) with significant effort and resource during the rebuttal to additionally evaluate our method for the stronger Llama series. Specifically, we train Llama-3.1-8B as the golden reward model. This model archives 75.8\% accuracy in the evaluation set. We then finetune Llama-3.1-1B on the preference data labeled by the golden preference model to compare DPO and Adaptive-IS with regularization parameter $\beta=0.1$. We present the performance improvement and regularization initial results in the Tables below. The results show a consistent observation for strong models, where Adaptive IS again achieve superior performance while maintaining lower KL budget.
> >
> >   * Additionally, while DPO shows early convergence phenomeneon, our methods shows consistent performance as the training progress in the large models cases. This shows the robustness of our algorithms in both weak and strong models cases.
> >
> >     * The table below shows the win-rate evolution as training progress with Llama-3.2-1b model.
> >
> >         | Method | Step-20096 | Step-40192 |Step-60288| Step-80384| Step-92820|
> >         | -------- | -------- | -------- | -------- | -------- | -------- |
> >         | DPO     |  0.664    | 0.699     | 0.648 | 0.617 | 0.625 |
> >         |Adaptive-IS DPO | 0.613 | 0.648 | 0.6875 | 0.664 | 0.7148 |
> >     * The table above shows the KL divergence evolution as training progress with Llama-3.2-1b model.
> >
> >         | Method | Step-20096 | Step-40192 |Step-60288| Step-80384| Step-92820|
> >         | -------- | -------- | -------- | -------- | -------- | -------- |
> >         | DPO     |  5.968    | 11.7     | 9.18 | 21.13 | 13.42 |
> >         |Adaptive-IS DPO | 4.627 | 11.303 | 16.01 | 12.46 | 12.25 |
> >
> >     * We will include this discussion in our camera-ready version of the paper.
> >
> > **IN CONCLUSION**, we believe that our work and our responses have already adressed all the issues mentioned by the Reviewer, with thorough and rigorous theoretical analysis and extensive empirical demonstrations, including evaluation on a stronger model, as the Reviewer suggested. We sincerely hope that this effort can be appreciated by the Reviewer, and that the Reviewer will revise the rating of the paper accordingly.

---

### Official Review · Reviewer_zEKh · 2024-10-31

**Soundness:** 3
**Presentation:** 3
**Contribution:** 2
**Rating:** 8
**Confidence:** 4

**Summary:**

This paper proposed to use adaptive importance sampling during offline post-training alignment of LLMs as a way to reduce over-optimization. They use a smoothed exponential IS estimator (where the exponent is the reciprocal of the length of the generation) in order to reduce the variance of the IS in exchange for some bias. Their experiments show that with this smoothed IS correction, they are able to reduce over-optimization in DPO and IPO and reach better performance in a lower KL budget in the TL;DR summarization task. They also show that distribution shift is indeed problematic and makes over-optimization worse.

**Strengths:**

-  The idea is simple and easy to integrate into existing algorithms like DPO and IPO as done in the paper.
-  The paper shows clear gains in terms of less overfitting and better performance per KL budget.

**Weaknesses:**

Overall, some more ablations or more in-depth investigation is lacking. There isn’t a good understanding of how important picking alpha is for the experiments. There is also not an investigation into how distribution shift (section 4.3) interacts with IS. See questions for more details.

**Questions:**

- Figure 1 and section 3.2 could be improved a lot by using the smoothed IS estimate and ablating over alpha, showing how the tradeoff between bias and variance works in such a toy domain as well.
- Some ablation over alpha in TL;DR would also be very insightful. Right now it seems that alpha is set arbitrarily to 1/|y|, when other values like 1/sqrt(|y|) might also be under consideration. Even if they perform worse, it is valuable insight into how alpha affects the performance or over-optimization.
- The distribution shift experiments in section 4.3, while does show that distribution shift is directly harmful, seems to be missing the natural followup of applying IS or smoothed IS in order to improve. How much does adding IS help? Or would it actually hurt performance because the data is no longer form pi_ref, so importance sampling towards pi_ref is actually increasing the distribution shift? Knowing something about how IS interacts with distribution shift would be a very good contribution to the paper.

---

> ### Author Response · Authors · 2024-11-22
>
> Thank you for your thoughtful review and valuable feedback. Below we address your concerns.
>
> **1. The motivation of adaptive smoothing factor**
>
> In auto-regressive language models, our importance weights are computed as the ratio of probabilities from two distributions, accumulated over all tokens. If the model’s distribution $\pi_\theta$ differs greatly from the reference distribution $\pi_{\text{ref}}$, the variance of importance sampling estimators can grow exponentially with the number of tokens in the output. To address this, we applied exponential smoothing of importance weights, using a smoothing factor $\alpha = \frac{1}{T}$. This adjustment reduces the variance significantly, preventing exponential growth with the response length. We provide a detailed analysis in the general response and in Sections E and F of the Appendix in our revised manuscript.
>
> **2. Ablation over alpha in TL;DR task**
>
> We conducted additional experiments where the value of $\alpha$ is fixed instead of adaptive (Section F, Appendix). The value of $\alpha$ are vary in $\{0.0, 0.05, 0.1, 0.2, 0.4 \}$ in the AIS-DPO loss function on Reddit TL;DR dataset.
> We observed that increasing $\alpha$ helps increase the regularization effect and win rate. However, up to a specific point (around 0.1), the regularization effect starts to diminish due to high variance in the importance ratio. Moreover, Adaptive-IS with $\alpha=1/T$ achieves the best result.
>
> We also provide an ablation of $\alpha =\frac{1}{\sqrt{|y|}}$. Although it achieves lower win-rate than DPO and AIS-DPO. We speculate that setting setting $\alpha=\frac{1}{\sqrt{y}}$ can still have high variance in the importance ratio, leading to a small number of samples having enormous weights can potentially dominate learning signals of other valuable samples [1].
> Square root IS-DPO still shows better regularization effect than other 2 objectives. Thus, the value of $\alpha$ can be used to control regularization effect in DPOs.
>
>
> **3. How does AIS perform in Section 4.3 and when the preference dataset is not generated from the SFT policy**
>
> We presented new results of applying IS on the distribution shift experiment in Section G, Appendix. Our observations are: (1) when the data distribution is close to $\pi_{\text{ref}}$, Adaptive IS and DPO show similar performance in terms of win rate and KL divergence, but both still suffer from the distribution shift effect, and (2) as the data distribution shifts away from $\pi_{\text{ref}}$, AIS-DPO is shown to achieve better regularization and win rate compared to standard DPO even when the data is no longer from $\pi_{\text{ref}}$. This phenomenon is helpful in practice, where the preference data is usually generated from an unknown policy $\mu$, not from $\pi_{\text{ref}}$. AIS-DPO can still improve performance and regularization when $\pi_{\text{ref}}$ is not far from $\mu$.
>
> We would like to thank the reviewer again for your thoughtful reviews and valuable feedback. We would appreciate it if you could let us know if our responses have addressed your concerns and whether you still have any other questions about our rebuttal.
>
> [1] Is Value Learning Really the Main Bottleneck in Offline RL?

---

> > ### Comment · Reviewer_zEKh · 2024-11-22
> > **Thank you for your additional results**
> >
> > Thank you for running more ablations and experiments. In light of the new results, they greatly add to the analysis of the paper, and so I've raised my score.

---

> ### Author Response · Authors · 2024-11-25
> **Thank you for your positive response!**
>
> Thank you once again for your valuable feedback. We are pleased that our responses addressed your concerns and appreciate your support for the paper.

---

### Official Review · Reviewer_AdMV · 2024-11-04

**Soundness:** 3
**Presentation:** 2
**Contribution:** 2
**Rating:** 3
**Confidence:** 4

**Summary:**

The paper primarily deals with the issue of reward over-optimization in specifically Direct alignment algorithms and proposes an adaptive low variance importance sampling strategy to mitigate the issue, with an exponential smoothing technique that balances bias and variance in IS estimates. The proposed method effectively reduces over-optimization by achieving higher model win rates and maintaining a lower KL divergence budget than baselines.

**Strengths:**

The paper primarily deals with the issue of reward over-optimization in specifically Direct alignment algorithms. The over-optimization issue is an extremely critical concern in the current alignment paradigms, and arises due to a distributional shift between offline training data and the LM's current policy, leading to increased probability on out-of-distribution (OOD) responses. The paper introduces an adaptive importance sampling strategy to mitigate the distributional shift issue using an exponential smoothing technique that balances bias and variance in IS estimates.

**Weaknesses:**

1. The importance sampling term defined in the equation in line 212, suggest that the original equation is E_{\pi_{\theta}}[\rho_theta]? Can you mathematically show why thats the case? In the context of online RLHF, it makes sense as shown in [1], but in offline whats the exact ideal optimization objective, leading to this importance weight? Can you specify, will be helpful. Also, highlight the difference from [1].
2. Whats the mathematical motivation behind choosing the value of the alpha? How does it affect the convergence?
3. There are several works on pessimism based methods to achieve reward over-optimization which are similar in principles, hence its not clear the novelty of the proposed work. A detailed comparison and contrast is critical to understand the novelty of the proposed approach.

References:

[1]. Sail: Self-improving efficient online alignment of large language models

[2]. Iterative data smoothing: Mitigating reward overfitting and overoptimization in rlhf

[3]. Provably Mitigating Overoptimization in RLHF: Your SFT Loss is Implicitly an Adversarial Regularizer

[4]. Provably Mitigating Overoptimization in RLHF: Your SFT Loss is Implicitly an Adversarial Regularizer

**Questions:**

The Taylor expansion is shown around rho_theta = 0, whats the point of expanding around  rho_theta = 0? It occurs when y_w = y_l or the preferred and chosen responses are very similar? Whats the intution behind expanding at that point is not clear.

---

> ### Author Response · Authors · 2024-11-22
>
> We appreciate your constructive comments on our paper. Please find our response to your comments below.
>
> **1. Why do we need to estimate $E_{\pi_{\theta}}[\rho_\theta]$? What is the ideal optimization objective?**
> Our motivation to derive the original equation comes from the fact that DAAs can be derived from vanilla policy gradient (VPG) ([4], [5]), an on-policy algorithm. We have shown the equivalence in Section H, Appendix, where DAAs can be seen as maximizing binarized reward using policy gradient. However, this equivalence only holds when we consider the online version of DPO or IPO. In off-policy setups, DAAs can suffer from the distribution shift problem, which has been well-studied in offline RL literature [3]. This also explains the ineffectiveness of regularization in DAAs when using offline data due to the sampling bias in the regularization objective [9, 10].
>
> A method to reduce the effect of sampling bias is using importance sampling to estimate the loss function under $\pi_\theta$  distribution given samples from a reference distribution $\pi_{\text{ref}}$ [3, 11]. This leads to importance weights that we proposed in the paper.

---

> > ### Author Response · Authors · 2024-11-22
> >
> > **2. Differences between SAIL and AIS-DPO**
> > While both our approach and SAIL addressing distribution shift problem in RLHF. Our motivation come from deriving DAAs methods as policy gradient approach, which might suffer from distribution shift problem in offline setting as widely studied in offline RL literature [3].
> >
> > SAIL formulating Online RLHF as bilevel optimization problem, which showing that standard RLHF suffer from distribution shift problem in the reward learning phase. They then reduce to single level objective by utilizing the equivalence between the reward function and the LLM policy [4] to reduce to a single level objective:
> >
> > $$
> > \max\_{\pi\_\theta}J (\pi\_\theta) = \mathbb{E}\_{x\sim \mathcal P, y\_i\sim\pi\_\theta(\cdot|x), (y\_w\succ y\_l)\sim p^*}\left[\log\sigma\left(\beta\log\frac{\pi\_\theta(y\_w|x)}{\pi\_{\text{ref}}(y\_w|x)}-\beta\log\frac{\pi\_\theta(y\_l|x)}{\pi\_{\text{ref}}(y\_l|x)}\right)\right]
> > $$
> >
> > Although our derivation from Policy Gradient lead to the same maximization objective as SAIL. There still exists some differences between SAIL and our approach:
> >
> > **Gradient Evaluation:** While we both share the similar objective, SAIL gradient can be express as the sum of two gradient terms:
> >
> > $$
> > \nabla\_{\theta} J(\theta) =  \sum\_{x, y\_w, y\_l} \nabla\_{\theta} \hat{\pi}\_{\theta} (y\_w, y\_l |x) F\_{\theta} (x , y\_w, y\_l)  +  \mathbb{E}\_{[\mathbf x\sim \mathcal{P}, y\_i \sim \pi^*\_{r(\cdot|\mathbf{x})}, (\mathbf y\_w \succ \mathbf y\_l)\sim p*]}[\nabla\_{\theta} [F\_{\theta} (x, y\_w, y\_l)]
> > $$
> >
> > From our policy gradient derivation, we update the policy using only the second gradient term, omitting the gradient from the trajectory distribution (the first gradient term). As stated in the paper, this novel first gradient term guides the policy $\pi\_\theta$  to generate $y\_w$ and $y\_l$ with maximum diversity by maximizing the log preference probability, therefore ensuring efficient exploration during sampling.
> >
> > It is unclear how this approach can be applied to offline cases, as it require model generated samples and how it can address distribution shift in DAAs, since the first gradient terms only enhance diversity during sampling, which is prohibited in DAAs. Notably, an online variant of DAAs [5], which approximate only the second gradient term (similar to our approach), has shown to mitigate reward-overoptimization problem and can even outperform other online alignment algorithms such as RLHF, RLAIF without the addition of the first gradient term.
> >
> > **Differences between Online and Offline Setup:** SAIL operates in an Online Iterative RLHF setup, allowing the use of self-generated samples to estimate the objective. In contrast, our work focuses on the Offline setup, where self-generated samples are prohibited. We propose using importance sampling to correct sampling bias in the off-policy setting without requiring model-generated samples.
> >
> > Although SAIL proposes an offline variant, SAIL-DP, by adding a gradient term that leverages the preference probability of the current LM policy $\pi_\theta$ for self-improvement, this additional term does not address the distribution shift problem in offline cases. Moreover, their experimental results show "over-optimization behavior" in SAIL-DP, achieving larger reward margins in the offline dataset but lower win rates compared to other online variants.
> >
> >
> >
> > ----
> >     [1] Contrastive Policy Gradient: Aligning LLMs on sequence-level scores in a supervised-friendly fashion
> >     [2] Pairwise Proximal Policy Optimization: Harnessing Relative Feedback for LLM Alignment
> >     [3] Offline Reinforcement Learning: Tutorial, Review, and Perspectives on Open Problems
> >     [4] Direct Preference Optimization: Your Language Model is Secretly a Reward Model
> >     [5] Direct Language Model Alignment from Online AI Feedback
> >     [6] Reinforcement learning: An introduction
> >     [7] Policy gradient methods for reinforcement learning with function approximation
> >     [8] SAIL: Self-Improving Efficient Online Alignment of Large Language Models
> >     [9] Generalized Preference Optimization: A Unified Approach to Offline Alignment
> >     [10] Scaling Laws for Reward Model Overoptimization in Direct Alignment Algorithms
> >     [11] Notes on importance sampling and policy gradient
> > ----

---

> > > ### Author Response · Authors · 2024-11-22
> > >
> > > **3. The mathematical motivation behind choosing the value of the alpha**
> > >
> > > Our motivation for setting $\alpha=1/T$ is to recude the variance of importance weighted estimator. Since we are working with an auto-regressive language models, the importance weights are computed as the product of the importance ratio of many timesteps i.e.
> > > \begin{equation*}
> > >     w(\mathbf{x},\mathbf{y}) =\frac{\pi\_\theta(\mathbf{y}|\mathbf{x})}{\pi\_{\text{ref}}(\mathbf{y}|\mathbf{x})}=\prod\_{t=1}^T\frac{\pi\_\theta(\mathbf{y}\_t|\mathbf{x},\mathbf{y}\_{<t})}{\pi\_{\text{ref}}(\mathbf{y}\_t|\mathbf{x},\mathbf{y}\_{<t})}
> > > \end{equation*}
> > >
> > > Thus, the variance of the IS estimator accumulates multiplicative.
> > > For instance, we analyze a setting where the reference mode $\pi\_{\text{ref}}$ is a uniform distribution over the vocabulary space $V$. The importance weight in this setting is given by the following equation.
> > > \begin{equation*}
> > >     w(\mathbf{x},\mathbf{y}) = |V|^T\prod\_{t=1}^T\pi\_\theta(\mathbf{y}\_t|\mathbf{x},\mathbf{y}\_{<t})
> > > \end{equation*}
> > > The variance of the importance weights can grow exponentially large with respect to the number of tokens in the response $\mathbf{y}$.
> > >
> > > \begin{align*}
> > > \underset{{\mathbf{y} \sim \pi\_{\text{ref}}(\cdot|\mathbf{x})}}{\operatorname{Var}}\left[w(\mathbf{x},\mathbf{y})\right]&=|V|^{2T}\underset{{\mathbf{y} \sim \pi\_{\text{ref}}(\cdot|\mathbf{x})}}{\operatorname{Var}}\left[\prod\_{t=1}^T\pi\_\theta(\mathbf{y}\_t|\mathbf{x},\mathbf{y}\_{<t})\right].
> > > \end{align*}
> > > By using exponential smoothing importance weights
> > > \begin{align*}
> > > w(\mathbf{x},\mathbf{y}) = |V| \prod\_{t=1}^T\pi\_\theta(\mathbf{y}\_t|\mathbf{x},\mathbf{y}\_{<t})^\alpha
> > > \end{align*}
> > > and choosing the value of $\alpha=\frac{1}{T}$, the variance of the importance weights is reduced significantly and does not grow exponentially with respect to the number of tokens in the response $\mathbf{y}$.
> > > \begin{align*}
> > > \operatorname{Var}[w(\mathbf{x},\mathbf{y})]&=|V|^{2}\operatorname{Var}\left[\pi\_\theta(\mathbf{y}|\mathbf{x})^\alpha\right]
> > > \end{align*}

---

> > > > ### Author Response · Authors · 2024-11-22
> > > >
> > > > **4. How does the value of alpha affect the convergence?**
> > > >
> > > > We present a new figure on training loss of AIS-DPO with different $\alpha$ values in Section F, Appendix. We observed that the loss is stable in all settings and larger values of $\alpha$ lead to faster convergence than smaller values.
> > > >
> > > > **5. There are several works on pessimism based methods to achieve reward over-optimization which are similar in principles, hence its not clear the novelty of the proposed work. A detailed comparison and contrast is critical to understand the novelty of the proposed approach.**
> > > >
> > > > Iterative Data Smoothing (IDS) [4] shows reward over-optimization happens in standard RLHF due to limited preference data and noisy labels when training the reward model with standard cross-entropy objective. They propose a pessimistic MLE objective, which learns the ground truth reward for samples that are compared enough times and ignores infrequently covered comparisons.
> > > >
> > > > While their approach works in the standard RLHF setting, we focus on offline alignment, which doesn't require an explicit reward model. It's unclear how we can apply IDS to the offline setting for the following reasons:
> > > >
> > > > - **The implicit reward in DAAs is less accurate than the classifier reward model:** Previous works [1], [2] have shown that classifier reward models can achieve 10-20% higher accuracy than parameterizing reward as an LLM policy $\pi_\theta$. This lower predictive accuracy means updating new labels $y$ as a mixture of previous values and newly predicted probabilities risks introducing noisy labels. Moreover, there is a trade-off between fitting the offline preference data (achieving high accuracy) and deviating from the reference model. To achieve high accuracy, the LLM must deviate from the reference model, increasing the chances of reward over-optimization. (see [2])
> > > > - **Classification accuracy doesn't necessarily predict generative performance:** While IDS achieves higher accuracy than the standard MLE objective, this may not translate to improved generative performance in DAAs, as shown in [1].
> > > > - **Reward Over-optimization occurs even before an epoch has elapsed:** While IDS is applied after each $t$-th epoch, [3] has pointed out that reward over-optimization in DAAs methods can happen before a single epoch is complete. This raises questions about the optimal frequency of applying IDS for regularization.
> > > >
> > > > In contrast, our paper shows that reward over-optimization happens due to the mismatch between offline data distribution and the current LLM policy. We then proposed an adaptive importance sampling approach to minimize this distribution gap to make offline learning more effective and sample efficient by incorporating importance weights to estimate Online-DAAs objective.
> > > >
> > > > Regularized Preference Optimization (RPO) ([5]) also shows reward-overoptimization happens due to distribution shift problem, similar to ours. They propose a theoretical algorithm that minimizes the DPO loss and a SFT term to mitigate reward over-optimization. RPO ensures alignment with the baseline policy to stabilize training. On the other hand, we propose to mitigate the distribution shift problem by adding an importance ratio to estimate samples under the current LM policy $\pi_\theta$.
> > > >
> > > > We have provided a detailed comparison with RPO in section I. As expected, DPO achieved the lowest performance with a higher KL divergence than RPO and Adaptive-IS DPO. Adaptive-IS DPO was able to show superior performance than RPO under similar KL budget without requiring any additional hyper-parameters.
> > > >
> > > > **6. The intuition behind expanding at $\rho_\theta = 0$.**
> > > >
> > > > As we typically initialize $\pi_\theta =\pi_{\text{ref}}$ at the beginning of training in DAAs, the log ratio differences $\rho_\theta=\log\frac{\pi_\theta(y_w)}{\pi_{\text{ref}}(y_w)} - \log\frac{\pi_\theta(y_l)}{\pi_{\text{ref}}(y_l)}$ start at zero. This motivates us to consider a Taylor expansion of DAAs’ loss functions around $\rho_\theta = 0$. This approximation holds when $\rho_\theta$ is small—in other words when the LM policy $\pi_\theta$ remains close to the reference model.
> > > >
> > > > We hope we have cleared your concerns about our work. We have also revised our manuscript according to your comments, and we would appreciate it if we could get your further feedback.
> > > >
> > > > ----
> > > >     [1] Understanding the performance gap between online and offline alignment algorithms
> > > >     [2] Preference Learning Algorithms Do Not Learn Preference Rankings
> > > >     [3] Disentangling Length from Quality in Direct Preference Optimization
> > > >     [4] Iterative Data Smoothing: Mitigating Reward Overfitting and Overoptimization in RLHF
> > > >     [5] Provably Mitigating Overoptimization in RLHF: Your SFT Loss is Implicitly an Adversarial Regularizer
> > > > ----

---

> > > > > ### Comment · Reviewer_AdMV · 2024-11-25
> > > > > **Response to Rebuttal**
> > > > >
> > > > > Thanks for the detailed response and providing clarifications to the other references. I appreciate the details provided in the rebuttal which improves the clarity.
> > > > >
> > > > > However, few concerns remains -
> > > > > 1. So this method is inherently offline and not online, so importance sampling is just for reducing the variance, right?
> > > > > 2. The reward used is dependent on the policy generating the response? So, its dependent on $\theta$ right? In line 977, while taking the derivative that needs to be considered as well right? Can you please derive this considering the reward being considered in the paper and showing what the actual optimization problem is, its not clear still.
> > > > > 3. Can you provide detailed explaination on expanding $\rho_{\theta} = 0$ since, it doesn't only refers when $\pi_{\theta}$ and $\pi_{\text{ref}}$ are close or far, but also when $\pi_{\theta}(y_w|x)$ and $\pi_{\theta}(y_l|x)$ are far, this won't hold right?

---

> ### Author Response · Authors · 2024-11-27
>
> We thank the reviewer for their feedback and will respond to the raised questions below.
>
> **1. This method is inherently offline and not online, so importance sampling is just for reducing the variance, right?**
>
> In this work, we follow the offline settings presented in [3,1], where only a fixed dataset is provided; we do not rely on model-generated samples or external rewards for learning.
>
> The role of our importance sampling approach does not actually aim to reduce the variance but to adjust the distributional differences between the offline data and the current policy, for better approximation in the online alignment algorithms.
> Specifically, importance sampling will reweight samples from the offline data such that those with high likelihood under the current learned model, $\pi_\theta$, are given more "weights", and those with low likelihood will be down-weighted. This helps to prevent over-optimizing samples that already have low-likelihood under $\pi_\theta$.
>
>
> **2. The reward used is dependent on the policy generating the response? So, its dependent on $\theta$
>  right? In line 977, while taking the derivative that needs to be considered as well right? Can you please derive this considering the reward being considered in the paper and showing what the actual optimization problem is, its not clear still.**
>
>
> In standard RL, the reward is a fixed function that assigns a scalar reward to each state-action pair and it does not depend on the policy parameters; thus the gradient of the reward with respect to the policy parameters will be zero.
> In RLHF, there exists an additional KL divergence term $\text{KL}(\pi_\theta||\pi_{\text{ref}})$. One can define the additional KL regularization as maximizing the new reward $r(x,y)$:
> $$\mathbb E_{y\sim\pi_\theta(\cdot|x)}\left[r(x,y)\right]=\mathbb E_{y\sim\pi_\theta(\cdot|x)}\left[r_\phi(x,y)-\beta\log\frac{\pi_\theta(y|x)}{\pi_{\text{ref}}(y|x)}\right]$$
>
> Although this reward depends on the policy's parameters due to the addition of $-\log\pi_\theta(y|x)$ term, the gradient of this new reward w.r.t the policy's parameters will still vanish using the log-derivative trick ([6]):
> $$
> \begin{align*}
> &\mathbb E_{y\sim\pi_\theta(\cdot|x)}\left[\nabla_\theta r(x,y)\right  ]\\\\
> &= \beta\mathbb E_{y\sim\pi_\theta(\cdot|x)}\left[-\nabla_\theta\log\pi_\theta(y|x)\right]\\\\
> &= 0
> \end{align*}
> $$
> The **property** results show that if we replace the reward in policy gradient objective with a binary signal, which depends on which generation is preferred, we follow the same gradient as IPO. In this sense, policy gradient also subsumes direct alignment approaches (see in [5]).
>
>
> **3. Can you provide detailed explaination on expanding $\rho_\theta=0$ since, it doesn't only refers when $\pi_\theta$ and $\pi_{\text{ref}}$ are close or far, but also when $\pi_\theta(y_x|x)$ and $\pi_\theta(y_l|x)$ are far, this won't hold right?**
>
>
> Note that the log-ratio difference $\rho_\theta=\log\frac{\pi_\theta(y_w|x)}{\pi_{\text{ref}}(y_w|x)}-\log\frac{\pi_\theta(y_l|x)}{\pi_{\text{ref}}(y_l|x)}$ depends on both $\pi_\theta$ and $\pi_{\text{ref}}$. If $\pi_\theta(y_w|x)$ and $\pi_\theta(y_l|x)$ are far apart (i.e the difference is high), when the reference model is close to $\pi_\theta$, it would similarly exhibit a large difference between $\pi_{\text{ref}}(y_w|x)$ and $\pi_{\text{ref}}(y_l|x)$. This would ensure that the log-ratio difference $\rho_\theta$ remains small, even when $\pi_\theta(y_w|x)$ and $\pi_\theta(y_l|x)$ are far apart.
>
> We have provided a detailed explanation of expanding $\rho_\theta=0$ in Section J of the Appendix. Our analysis indicates that during training, there is a mismatch in data distribution: KL divergence is computed based on samples from the current LM, while Offline regularization in DAAs is calculated based on offline samples. This can result in cases where minimizing offline regularization objectives might not necessarily reduce KL divergence. The reason is that the offline samples may not accurately reflect those produced by the current policy, which can potentially lead to performance degradation (see [2], [4]).
>
>
> **In summary,** we have answered the questions posed by the reviewers. We hope that these questions clarify all the reviewer's concerns about our work. Nevertheless, if there are any additional questions, please kindly let us know during the rebuttal period and we will promptly provide our responses.
>
> ----
>     [1] Understanding the performance gap between online and offline alignment algorithms.
>     [2] Generalized Preference Optimization: A Unified Approach to Offline Alignment
>     [3] Scaling Laws for Reward Model Overoptimization in Direct Alignment Algorithms
>     [4] Preference Learning Algorithms Do Not Learn Preference Rankings
>     [5] Contrastive Policy Gradient: Aligning LLMs on sequence-level scores in a supervised-friendly fashion
>     [6] Notes on policy gradients and the log derivative trick for reinforcement learning
>
> ----

---

> > ### Author Response · Authors · 2024-12-01
> >
> > Dear Reviewer AdMV,
> >
> > As the rebuttal period is **coming to an end very shortly**, we again thank the Reviewer for the valuable initial comments and the additional questions, which we had previously responded during the rebuttal phase; all the corresponding changes are now incorporated in the revised paper during rebuttal.
> >
> > We also hope that the reviewer take into account our effort in the rebuttal phase and the significance of our work, including the novel idea that can be integrated into any existing DAAs and our contributions in analyzing this idea both theoretically and empirically, in addressing the overfitting problem of existing DAAs. Consequently, we kindly ask the reviewer to reconsider the original rating and raise the score accordingly.
> >
> > Sincerely,
> > Paper 14149’s Authors

---

### Official Review · Reviewer_u6Dx · 2024-11-04

**Soundness:** 3
**Presentation:** 3
**Contribution:** 2
**Rating:** 6
**Confidence:** 4

**Summary:**

This work addresses the reward optimization problem in direct alignment algorithms (DAAs) from the angle of distribution shift. In existing DAAs, the KL estimation is only unbiased when the samples are on-policy. However, as the policy being updated during learning, the responses from the offline dataset become off-policy, and thus distribution shift happens.

To address this issue, the authors propose adaptive importance sampling (AIS) as a solution. Assuming the preference data are generated from the SFT policy, AIS applies an importance sampling weight on each data point to correct the off-policyness. This weight term is further adapted by an exponential coefficient which is the inverse of the response length to tradeoff the bias and the variance. AIS is first evaluated in a toy example and demonstrates better estimation of the KL divergence than its unweighted counterpart. When combined with DPO, AIS demonstrates better KL-win rate tradeoff and higher peak performance than the baseline in a simulated setup, following Gao _et al_, 2022. The authors also conducted some empirical analysis in the simulation setup to understand the detriment of distribution shift.

**Strengths:**

This work addresses a widely observed phenomenon where DAAs like DPO suffers from reward overoptimization even before completing the first epoch of the dataset. Insights into this phenomenon can help us understand the underlying mechanism of DAAs  and resolving this issue can mitigate the gap between online and offline algorithms and can provide us with computationally cheap yet performant alignment algorithms.

The proposed solution, AIS, is a principled algorithm with well-understood theoretical grounding. Empirically, AIS demonstrates more effective KL regularization and strong performance over the baseline. AIS is simple, easy to implement, and preserves the low computational cost of offline DAAs.

In terms of presentation, overall the paper is easy to follow. The work is well motivated and the method is clearly explained. The authors did a good job connecting to existing works in the literature.

**Weaknesses:**

Important analysis on the proposed AIS method is missing. Importance sampling is one of the simplest techniques to address off-policy learning. The authors claim that vanilla IS suffer from high variance and thus an adaptive heuristic is applied to make a tradeoff between the bias and the variance. However, there is no analysis into this adaptive heuristic to justify its necessity and to provide insights into how this tradeoff impacts the overall performance.

Similarly, the empirical analysis in Section 4.3 demonstrates the detriment of distribution shift to DAAs. One natural question to ask is, as a method proposed for addressing distribution shift, how does AIS perform in these experiments? The current study does not provide any results to answer this question.

One limitation the authors did not call out in the limitation section is that AIS assumes that the preference dataset is generated from the SFT policy. However, this is not always the case in practice. Usually the responses in the preference dataset are sampled from different generations of the same data class, or even from different model classes. Thus this assumption is often violated and it hinders the effectiveness of AIS.

Presentation-wise, the authors use inconsistent / incorrect citation formats through the paper. Calandriello _et al_ '24 should be cited in Section 3.1 for online DDAs. There are a few typos in writing. I think it should be "budget" in the last sentence of the abstraction. The Azar '23 and Gheshlaghi Azar '24 citations are citing the same paper.

## References
Calandriello _et al_ '24, Human Alignment of Large Language Models through Online Preference Optimisation, ICML 2024.

**Questions:**

1. In Figure 2, Figure 4, and Figure 5, how was the KL computed? Was it estimated by taking on-policy samples from the current LM?

2. Could the authors provide online alignment algorithm results such as PPO in the main evaluation in Section 4.2? In my opinion, AIS does not need to outperform PPO. The main purpose is to provide better context to the readers. These results can help the readers understand how much of the online-offline gap can be explained by distribution shift and can be addressed by AIS. These results can also provide guidance for follow-up work and future research.

---

> ### Author Response · Authors · 2024-11-22
>
> Thank you for your review and positive feedback on our paper. We appreciate your acknowledgment of the clarity and organization of our work.
>
> Please find our response to your comments below.
>
> **1. How was the KL computed? Was it estimated by taking on-policy samples from the current LM?**
> The calculation of KL divergence in our paper is based on [1] and the KL is estimated by taking on-policy samples from the current LM $\pi_\theta$.
>
> Specifically, we first sample $N$ input prompts $\\{\mathbf{x}\_i\\}\_{i = 1}^N$ from the evaluation set. For each input prompt $\mathbf{x}\_i$, we generate a response $\mathbf{y}\_i$ using the current policy $\pi\_\theta$. Let $T\_i$ be the length of the response $\mathbf{y}\_i$, we compute the KL divergence between $\pi_\theta$ and $\pi_{\text{ref}}$ as follows:
>
> $$
> \frac{1}{N}\sum\limits\_{n=1}^N\sum\limits\_{t=1}^T\mathbb{KL}\big(\pi\_\theta(\cdot|\mathbf{x,y}\_{<t}), \pi\_{\text{ref}}(\cdot|\mathbf{x,y}\_{<t})\Big)
> $$
>
>
> We set $N=512$ in our experiments. For clarity, we have included the calculation of KL divergence between $\pi_\theta$ and $\pi_{\text{ref}}$ in the appendix.
>
> **2. What is the result of online alignment algorithm such as PPO in the main evaluation in Section 4.2? How much of the online-offline gap can be explained by distribution shift and can be addressed by AIS?**
>
> We agree that answering this question could provide greater insight. We conducted further experiments where we compare DAAs with online alignment methods. As PPO requries extensive computational resources, we consider an alternatives online algorithms which is REINFORCE Leave-One-Out (RLOO) [2]. Detail training setting and a figure presenting the result can be found in our updated manuscript (Section C in the Appendix). As expected, RLOO achives better win-rate compared to DPO and AIS-DPO and utilizing better KL budget. The result also shows that AIS-DPO help close the gap between offline and online algorithms.
>
> **3. The necessity of the adaptive heuristic**
>
> In auto-regressive language models, our importance weights are computed as the ratio of probabilities from two distributions, accumulated over all tokens. If the model’s distribution $\pi_\theta$ differs greatly from the reference distribution $\pi_{\text{ref}}$, the variance of importance sampling estimators can grow exponentially with the number of tokens in the output. To address this, we applied exponential smoothing of importance weights, using a smoothing factor $\alpha = \frac{1}{T}$. This adjustment reduces the variance significantly, preventing exponential growth with the response length. We provide a detailed analysis in the general response and in Sections E and F of the Appendix in our revised manuscript.
>
> **4. How does AIS perform in the synthetic distribution shift experiment and when the preference dataset is not generated from the SFT policy?**
>
> We presented new results of applying IS on the distribution shift experiment in Section G, Appendix. Our observations are: (1) when the data distribution is close to $\pi_{\text{ref}}$, Adaptive IS and DPO show similar performance in terms of win rate and KL divergence, but both still suffer from the distribution shift effect, and (2) as the data distribution shifts away from $\pi_{\text{ref}}$, AIS-DPO is shown to achieve better regularization and win rate compared to standard DPO even when the data is no longer from $\pi_{\text{ref}}$. This phenomenon is helpful in practice, where the preference data is usually generated from an unknown policy $\mu$, not from $\pi_{\text{ref}}$. AIS-DPO can still improve performance and regularization when $\pi_{\text{ref}}$ is not far from $\mu$.
>
> **6. Incorrect citation formats and typos in writting**
> Thank you for these remarks. We corrected citations and typos in our revised manuscript.
>
> We hope that our response addresses the issues you raised. Please let us know if you have any additional concerns or questions.
>
>
> ----
>     [1] Understanding the performance gap between online and offline alignment algorithms.
>     [2] Back to Basics: Revisiting REINFORCE Style Optimization for Learning from Human Feedback in LLMs
> ----

---

> > ### Author Response · Authors · 2024-11-27
> >
> > We’d like to thank the reviewer again for the valuable insights in the comments. We believe that we have fully addressed the all the comments. Nevertheless, if there is any additional questions, we will be happy to promptly respond within the rebuttal period

---

> > > ### Author Response · Authors · 2024-12-01
> > >
> > > Dear Reviewer u6Dx,
> > >
> > > As the rebuttal period is **coming to an end very shortly**, we again thank the Reviewer for the valuable comments; we have incorporated all our responses to the revised paper accordingly. We also kindly ask the reviewer take into account our effort in the rebuttal phase and accordingly raise the rating of the paper.
> > >
> > > Sincerely,
> > > Paper 14149’s Authors

---

### Author Response · Authors · 2024-11-22
**General Response**

Dear AC and reviewers,

Thanks for your thoughtful reviews and valuable comments. We are encouraged by the endorsements that: (1) the main objective of this work, addressing over-optimization in DAAs, is well motivated (Reviewer u6Dx, Reviewer AdMV) (2) the proposed method, AIS, is principled, easy to implement, and preserves the low computational cost of offline DAAs (Reviewer u6Dx, Reviewer zEKh) (3) the effectiveness of AIS is demonstrated by experiment results (Reviewer zEKh, Reviewer u6Dx) (4) the paper is easy to follow (Reviewer WtSX, Reviewer u6Dx) and have thorough literature review (Reviewer u6Dx)


One of the common questions from reviewers is about the motivation behind choosing the value of the alpha and some ablations over alpha in the TL;DR task. We first address this comment here.

**1. The necessity of the adaptive heuristic**

Since we are working with auto-regressive language models, the importance weights are computed as the product of the importance ratio of many timesteps.
$$
w(\mathbf{x},\mathbf{y}) =\frac{\pi_\theta(y|x)}{\pi_{\text{ref}}(y|x)}=\prod_{t=1}^T\frac{\pi_\theta(y_t|x,y_{<t})}{\pi_{\text{ref}}(y_t|x,y_{<t})}
$$

Thus, the variance of the IS estimator accumulates multiplicatively. The variance can be really large if $\pi_\theta$ and $\pi_{\text{ref}}$ are very different. We analyze the setting where the reference mode $\pi_{\text{ref}}$ is a uniform distribution over the vocabulary space $V$ then the importance weight is given by
$$
w(x,y) = |V|^T\prod_{t=1}^T\pi_\theta(y_t|x,y_{<t}).
$$
The variance of the importance weights can grow exponentially large with respect to the number of tokens in the response $y$:

$$
\begin{align*}
\operatorname{Var}(w(x,y))&=|V|^{2T}\operatorname{Var}(\pi_\theta(y|x)).
\end{align*}
$$
By using exponential smoothing importance weights
\begin{align*}
w(x,y) = |V| \prod_{t=1}^T\pi_\theta(y_t|x,y_{<t})^\alpha
\end{align*}
and choosing the value of $\alpha=\frac{1}{T}$, the variance of the importance weights is reduced significantly and does not grow exponentially with respect to the number of tokens in the response $y$.
$$
\begin{align*}
\operatorname{Var}(w(x,y))&=|V|^{2}\operatorname{Var}(\pi_\theta(y|x)^\alpha)
\end{align*}
$$

We conducted additional experiments where the value of $\alpha$ is fixed instead of adaptive (Section F, Appendix). We observed that increasing $\alpha$ helps increase the regularization effect and win rate. However, up to a specific point (around 0.1), the regularization effect diminishes due to high variance in the importance ratio. Moreover, Adaptive-IS with $\alpha=1/T$ achieves the best result.

---
We are glad to answer any further questions you have on our submission.

---

### Author Response · Authors · 2024-11-25
**Any Questions from the Reviewers before the Rebuttal/Discussion Period Ends?**

Dear reviewers,

We would like to thank all reviewers again for providing constructive feedback and raising questions to help us improve the quality of our paper. We have addressed all the questions and suggestions from the reviewers in our rebuttal with new experiments and paper revisions (please see our revised submission) as recommended.

We would appreciate it if you could let us know if there are additional questions or concerns about our revision and rebuttal.

We would be happy to do any follow-up discussion or address any additional comments.

Best regards,

Authors

---

### Author Response · Authors · 2024-12-01
**Thank you for your valuable comments and welcome additional questions!**

Dear reviewers,

Thank you for your thoughtful review and active engagement during the Discussion period.

We have provided detailed responses addressing all the questions and suggestions from the reviewers in our rebuttal. We believe our work is practical in solving over-optimization problems in DAAs in training LLMs.

As the discussion phase comes to an end, we kindly ask if you could confirm your evaluation or share any additional feedback. We would be happy to address any further comments promptly.

Thank you once again for your valuable insights.

Best Regards,

Authors

---

### Author Response · Authors · 2024-12-04
**Summary of our responses and paper revision during the rebuttal phase**

Dear reviewers,

We again would like to thank all reviewers for their thoughtful comments.
We have obtained additional analysis and empirical results as well as additional discussion to demonstrate the effectiveness of our proposed method, most of which are now incorporated in the revised paper. For those that have yet been included in the revision, we will add them in the final version. Below is our summary of the major points during the rebuttal period:

1. We conducted additional experiments, including the evaluation on a stronger language model (LLama-3.2-1B), a larger language model (Pythia-2.8b), and a new baseline (Regularized Preference Optimization [1]). The results show that AIS can still consistently achieve favorable performance under these new experiments.
2. We provided a comparison of AIS with online alignment methods and demonstrated that AIS helps close the gap between online and offline alignment algorithms. (Section C, Appendix)
3. We provided detailed analysis and ablation studies to validate the use of adaptive smoothing factors in AIS. (Section E, F, Appendix)
4. We investigated AIS under a synthetic setting that simulate the scenario where the data is not sampled from the reference distribution $\pi_\text{ref}$. The result shows that AIS-DPO is shown to achieve better regularization and win rate compared to standard DPO even when the data is no longer from $\pi_{\text{ref}}$. (Section G, Appendix)
5. Furthermore, we clarify the experiment implementation (KL divergence and golden win-rate computation) and hyper-parameter settings to help reproduce our results. (Section B, D, Appendix)

We believe these points have helped strengthen our paper and we hope the reviewers will consider them in the final ratings of our work.

**References**

[1] Provably Mitigating Overoptimization in RLHF: Your SFT Loss is Implicitly an Adversarial Regularizer

---

### Meta-Review · Area_Chair_p3FD · 2024-12-22

**Metareview:**

This work proposes methods to mitigate the reward over-optimization issues in direct alignment algorithms. The algorithm proposed is a low variance importance weighted sampling strategy that is meant to minimize this issue. Unfortunately, some technical issues were raised by the reviewers. These are things such as the unbiasedness of the DPO gradient estimator and others.

**Additional Comments On Reviewer Discussion:**

After the discussion there was agreement among the reviewers about several technical issues. We encourage the authors to revise the manuscript and address these before resubmission.

---

### Decision · Program_Chairs · 2025-01-22

Reject